



# Impact of North Atlantic SST and Jet Stream anomalies on European Heat Waves

Julian Krüger[1], Robin Pilch Kedzierski[1], Karl Bumke[1], and Katja Matthes[1,2]

[1]Marine Meteorology Department, GEOMAR Helmholtz Centre for Ocean Research Kiel, Kiel, Germany.
[2]Faculty of Mathematics and Natural Sciences, Christian-Albrechts-Universität zu Kiel, Kiel, Germany.

**Correspondence:** Julian Krüger (jukrueger@geomar.de)

**Abstract.**

European heat waves have increased during the two recent decades. Particularly 2015 and 2018 were characterized by a widespread area of cold North Atlantic sea surface temperatures (SSTs) in early summer as well as positive surface temperature anomalies across large parts of the European continent during later summer. The European heat wave of 2018 is further
suggested to be induced by a quasi-stationary and high-amplified Rossby wave pattern associated with the so-called quasi-resonant amplification (QRA) mechanism. In this study, we evaluate the North Atlantic SST anomalies and the QRA theory as potential drivers for European heat waves for the first time in combination by using the ERA-5 reanalysis product.

A composite and correlation study reveals that cold North Atlantic SST anomalies in early summer favour a more undulating jet stream and a preferred trough-ridge pattern in the North Atlantic-European sector. Further we found that cold North Atlantic
SSTs promote a stronger double jet occurrence in this sector. Thus, favorite conditions for a QRA signature are evident together with a necessary preconditioning of a double jet. However, our wave analysis covering two-dimensional probability density distributions of phase speed and amplitude does not confirm a relationship between cold North Atlantic SSTs and the QRA theory, compositing cold SSTs, high double jet indices (DJIs) or both together. Instead, we can show that cold North Atlantic SST events enhance the dominance of transient waves. In the presence of a trough during cold North Atlantic events, we obtain
a slow-down of the transient waves, but not necessarily an amplification or stationarity. The deceleration of the transient waves result in a longer duration of a trough over the North Atlantic accompanied by a ridge downstream over Europe, triggering European heat episodes. Although a given DJI preconditioning may also be subject to the onset of certain QRA events, our study found no general relation between cold North Atlantic SST events and the QRA diagnostics.

Our study highlights the relevance of cold North Atlantic SSTs for the onset of high European temperatures by affecting
travelling jet stream undulations (but without involving QRA in general). Further attention should be drawn not only to the influence of North Atlantic SST year-to-year variability, but also to the effect of the North Atlantic warming hole as a negative SST anomaly in the long term, which is projected to evolve through climate change.



## 1 Introduction

During the two recent decades Europe experienced an increase in extreme weather events (Christidis et al., 2015; Coumou
et al., 2013). Several heat wave occurrences with severe impacts on society including an increase in mortality and infrastruc-
ture strain were observed during the $21^{st}$ century (Mcgregor et al., 2007). Several studies provide strong evidence that the
observed increase is due to the anthropogenically induced climate change (Coumou and Rahmstorf, 2012; Stott et al., 2016;
Diffenbaugh et al., 2017; Mann et al., 2018). A heat wave over central Europe in 2003 (Robine et al., 2008) and another over
Moscow in 2010 (Barriopedro et al., 2011) were responsible for 70,000 and 55,000 excess mortality, respectively. Another two
heat wave episodes affected particularly central Europe in 2015 (Dong et al., 2016) and Scandinavia in 2018 (Sinclair et al.,
2019).

But what are the drivers of such prolonged periods of anomalous heat? Several studies already tackled this issue, suggesting
that soil moisture anomalies and their interactions with the atmosphere through surface feedbacks including latent and sensible
heat fluxes strongly contribute to the development of heat waves (Black et al., 2004; Fischer et al., 2007). During the heat wave
of 2003 the soil moisture over Europe dried to a large extent due to strong radiative forcing. Subsequently the lack of soil
moisture was responsible for negative latent heat flux anomalies, which in turn led to a higher sensible heat flux conditioned
by extremely high surface temperatures (Sinclair et al., 2019).

Trouet et al. (2018) claimed that the observed increase in mid-latitude extreme weather events is in general related to upper-
tropospheric anomalies, associated with a narrow band of upper tropospheric winds, commonly known as the jet stream (Hall
et al., 2014). Duchez et al. (2016) suggested that a cold North Atlantic sea surface temperature (SST) anomaly could have
initiated a propagating Rossby wave train causing a stationary jet stream position that subsequently favoured the development
of high pressure and temperature extremes downstream over central Europe. In the study by Duchez et al. (2016), special atten-
tion is paid to the heat wave of 2015, confirming the relationship statistically: they observed an anti-correlation consisting of
negative North Atlantic SST anomalies and positive surface temperature anomalies over central Europe during boreal summer.

Another proposed mechanism for the generation of heat waves is the quasi-resonant amplification (QRA) mechanism, intro-
duced by Petoukhov et al. (2013) and applied by Kornhuber et al. (2017) to the analysis of heat waves. The QRA theory
provides a mechanism for the formation of persistent synoptic-scale wave patterns, inducing extreme weather events. A double
jet structure, with two distinct wind maxima over the mid-latitude upper-troposphere, is initially required for the onset of the
QRA mechanism (Kornhuber et al., 2017). The presence of a double jet over the North Atlantic sector is more likely during
periods of a positive Northern Annular Mode (NAM) compared to periods of a negative NAM (Tachibana et al., 2010). With
a double jet structure present, synoptic-scale waves are more likely to be trapped within mid-latitudes, without dissipating
meridionally (Wirth, 2020), therefore promoting resonance (Petoukhov et al., 2013; Coumou et al., 2014; Kornhuber et al.,
2017).

The study by Kornhuber et al. (2019) highlights the importance of the QRA mechanism for the summer 2018: the mid-latitude
circulation was characterized by anomalously persistent meandering patterns of the upper-tropospheric flow. A hemisphere-
wide wavenumber 7 pattern with an alternating system of troughs and ridges remained in the same longitudinal position for





several weeks. Persistent anticyclonic conditions over Europe were at least partly responsible for the heat wave occurrence in 2018. Kornhuber et al. (2019) noted that in composites of QRA events, a trough was present over the North Atlantic roughly around the region studied by Duchez et al. (2016). However no study has established a link between the two approaches so far.

Our study will attempt to bridge the work of Kornhuber et al. (2017) and Duchez et al. (2016) for the first time. The research objective of our study is the assessment of the North Atlantic SST influence on the occurrence of the double jet structure in the North Atlantic sector as well as its wave properties, which in turn affect European temperatures. Section 2 will present the data and methods used in our study. In section 3 we examine the effects of North Atlantic ocean anomalies on the jet stream and European surface temperatures by using composite and correlation analysis.

## 65  2   Data and methods

This study is based on the European Centre for Medium-Range Weather Forecasts (ECMWF) ERA5 reanalysis data (Hersbach et al., 2019). We use daily averages from 1979 to 2019. The 300hPa geopotential height and the 300hPa zonal and meridional wind component as well as the sea surface temperature (SST) and 2m-air temperature are studied. We use a horizontal resolution of 2.5° longitude × 2.5° latitude.

### 70  2.1   Double Jet Index

The investigation of double jet structures is crucial, because it provides a necessary precondition for the onset of the QRA-mechanism, which leads to Rossby-wave amplification and quasi-stationary conditions (Kornhuber et al., 2017). It is important to note, that our definition of the double jet differs from the study made by Kornhuber et al. (2017). Kornhuber et al. (2017) used a circumglobal zonally averaged zonal wind for double jet examination. We define a double jet index (DJI) only in the North

Atlantic sector for two main reasons: because a double jet predominantly forms within this longitude range and not globally (Hall et al., 2014), and because we want to focus on the effect on European temperatures. For this purpose, we produced bimonthly means of the climatological total wind speed of the 300hPa level for May and June as well as for July and August, respectively, visualized in Fig. 1. According to the climatological state, three individual branches are established, covering the poleward and equatorward maximum and the minimum in between. In consideration of the characteristic Southwest-Northeast

tilt of the jet stream in the Atlantic sector, we introduce a steplike detection with equal slopes throughout all branches and all months.

Averaged over the longitudinal range of 7.5° to 45° W, each branch is calculated for each month between May and August individually. For May and June the average is derived by using the values for the latitudinal ranges below:

- 47.5° - 55°N for the poleward maximum $U^{north}$

- 20° - 27.5°N for the equatorward maximum $U^{south}$

- 32.5° - 40°N for the minimum in between $U^{mid}$



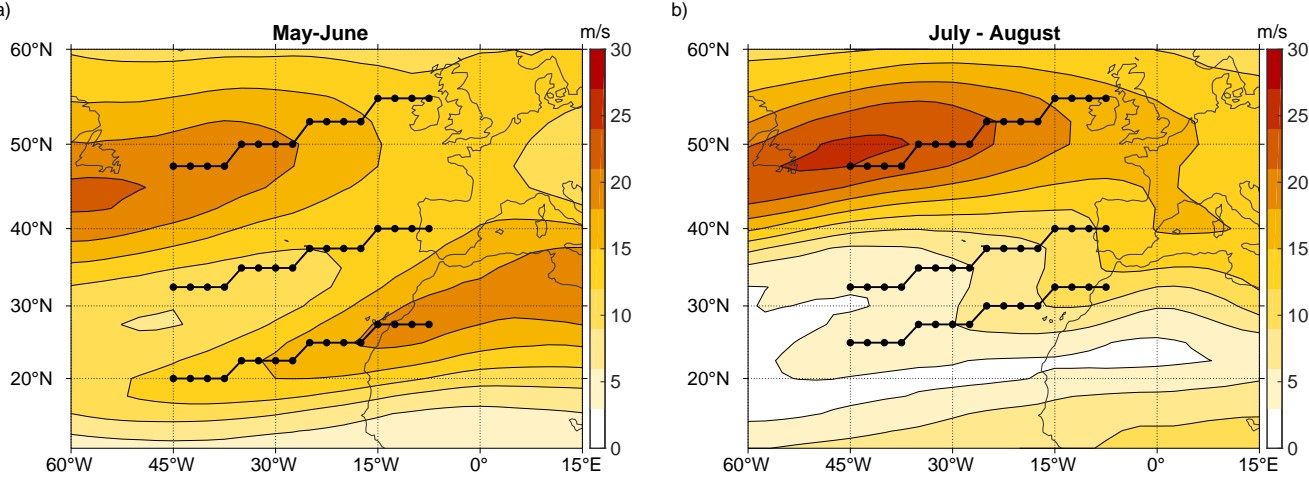

**Figure 1.** 300 hPa total wind speed of the climatological mean (1979-2019) for bimonthly averages (coloured) in ms$^{-1}$: a) May-June; b) July-August; solid connected grid points between 7 and 45°W are used for Double Jet Index calculation: a) for May and June; b) for July and August, respectively.

The seasonally varying structure of the jet stream climatology requires a latitudinal adjustment of the branches. For July and August the DJI is derived based on the same ranges except from the equatorward maximum, which is captured by a branch between 25° and 32.5°N. Note that, despite the difference in wind speeds, the position of the maxima-minima (solid-pointed

lines) in Fig 1 does not differ substantially, which is why bi-monthly averages are used instead of a more frequent one.

Averages for both maxima and the minimum in between are now applied for the calculation of the DJI, which mainly consists of the ratio between the DJI of an individual year t and the average of the DJI over all years (1979 - 2019, n = 41), where t = 1,..,n:

$$\mathrm{DJI}_t = \frac{U_t^{north} + U_t^{south} - U_t^{mid}}{\left[ \dfrac{\sum\limits_{t=1}^{n} U_t^{north} + U_t^{south} - U_t^{mid}}{n} \right]}.$$

**2.2 North Atlantic SST composites**

The influence of North Atlantic SST anomalies on the jet stream and on European surface temperatures is evaluated by using a composite study. After removing the seasonal variability of the SST data, the average over a North Atlantic box (45 - 60° N, 15 - 40° W) is created, coinciding with the area used for the analysis made by Duchez et al. (2016).

Afterwards monthly averages between 1979 and 2019 are produced. A lower and an upper quantile are selected, representing





cold and warm SST events. Additionally a reference is obtained by selecting the months between the 0.4 and 0.6 quantiles,
delineating the medium SST composite.

Cold, warm and medium composites were used for two different analysis:

1. The jet stream latitude over the North Atlantic-European sector: the latitude of the wind speed maximum within 45 and
90°N will be detected for each longitude between 90°E and 90°W. We apply a daily-based detection of the jet stream
maximum for all composites. Jet stream latitude anomalies are defined as the difference between warm and medium as
well as cold and medium SST events, respectively. Cold and warm SST composites are represented by the lower quantile
up to 0.25 and the upper quantile above 0.75, which is shown in Fig. 3a. Both composites are further produced for the
lower quantile of up to 0.1 and an upper quantile above 0.9, illustrated in Fig. S1a.

2. Spatial structures of geopotential height and surface temperature anomalies: we performed two sets of correlations: (i)
SST & 300 hPa GPH (ii) SST & 2m-air temperature. In both cases we correlate 15-day running means of the North
Atlantic SST average with the values of the second parameter on a daily basis for each grid point. Fig. 6 pictures the
composites with 0.25 and 0.75 quantiles. Composites with 0.1 and 0.9 quantiles are further shown in Fig. S2.

### 2.3    Wave diagnostics

In order to diagnose the wave characteristics and figure out whether heat waves originate from the QRA theory, we examine
two-dimensional probability density functions (PDFs) of phase speed and wave amplitude by using a Kernel-density estimate
similar to Kornhuber et al. (2017). Wave amplitude and phase of each wave number at each time step and latitude within the
selected range (37.5 - 57.5°N (Fig. 8, S4 (upper)) / 50 - 70°N (Fig. S3, S4 (lower))) were determined by applying a fast Fourier
transformation (FFT) of the meridional wind field at 300hPa with respect to longitude. The calculation of the phase speeds
deviates from the calculation made in the study by Kornhuber et al. (2017). Here, phase speed is calculated as the difference
between the phase transitions from one 12h time step to the next. Excluding a cubic-spline interpolation and doing a raw
difference instead, suggests a simple mean of the corresponding amplitude between the two neighboring 12h values, as the
phase speed is representative for the time in between the 12h interval.

For the definition of troughs and ridges in the North Atlantic we use 300hPa GPH. A zonal mean $z_{zonal}$ for the latitude band
45 - 60°N and another value $z_{Atl}$ for the North Atlantic box (14 - 40°W, 45 - 60°N) are both calculated and then combined as:

$$z_{anom} = \frac{z_{Atl} - z_{zonal}}{z_{zonal}} \cdot 100.$$

This anomaly $z_{anom}$ denotes the amplitude of troughs and ridges present relative to the zonal mean GPH. Fig. 9 shows the
distribution of trough-ridge occurrence with different SST quantiles over the North Atlantic.

We separate wave properties, comparing their kernel-density estimates when a trough or a ridge is present over the North
Atlantic, defined as days where $z_{anom} < 0.5$ and $z_{anom} > 0.5$, respectively. Based on a cold (0.1 quantile) or a warm SST event
(0.9 quantile) we perform the difference between troughs and ridges, shown in Fig. 8, S3 and S4.





# 3 Impact of North Atlantic SST anomalies

Low frequency variability associated with the North Atlantic ocean plays an important role for Northern Hemisphere climate (Kim et al., 2018). In this section implications of North Atlantic ocean anomalies are studied, particularly during boreal summer. In section 3.1 we focus on the investigation of the relationship with European surface temperatures and in section 3.2 on
jet stream properties. The causality of jet stream anomalies is analyzed using lead-lag correlations in section 3.3. Finally, an evaluation of wave parameters is performed in section 3.4.

## 3.1 Influence on European surface temperatures

Previous studies suggested that the Atlantic Ocean is an important driver of summertime climate over Europe (Sutton and Hodson, 2005; Cassou et al., 2005; Duchez et al., 2016). The North Atlantic Ocean was characterized by a widespread area of
negative anomalies during the summer 2015 and was crucial for the development of the European heat wave (Duchez et al., 2016; Mecking et al., 2019). We investigate whether a similar picture evolved during the heat wave in 2018. Therefore we

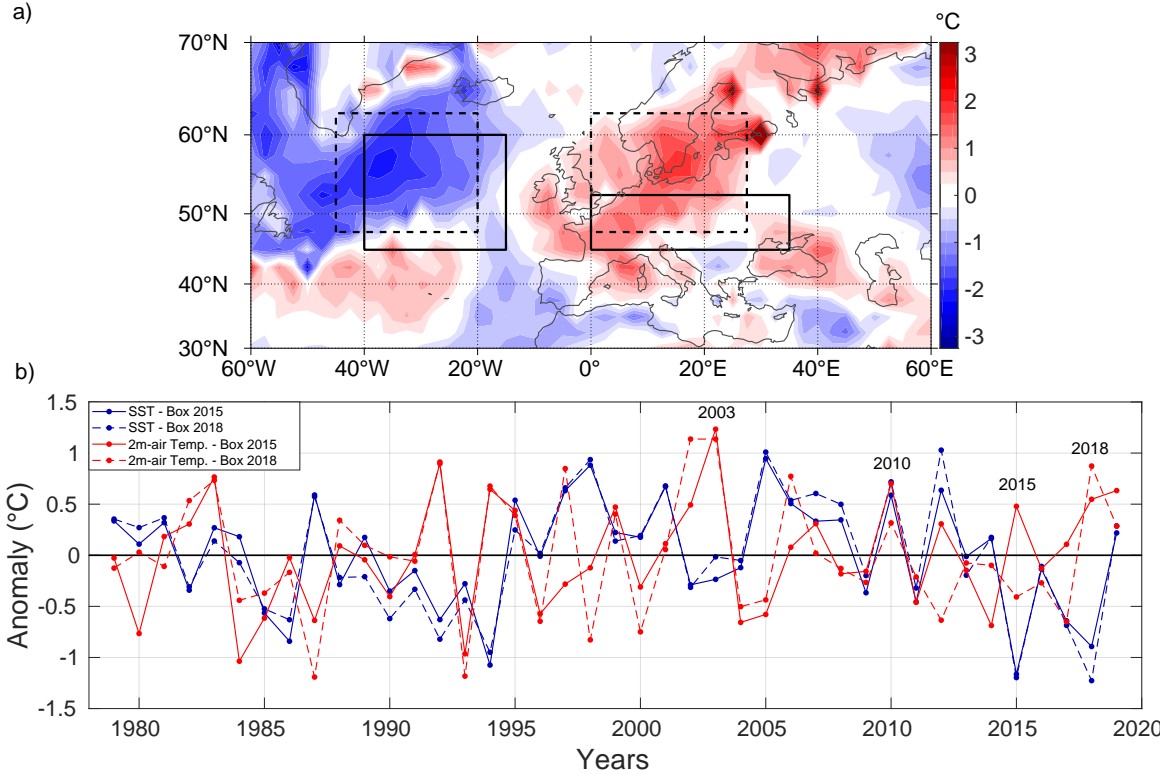

**Figure 2.** Detrended JJA anomalies for SST (ocean) and 2m-air temperature (land): a) 2018 anomalies; closed boxes are used for 2015 anomalies according to Duchez et al. (2016); dashed boxes indicates the area of largest 2018 anomalies; b) seasonal means (JJA) between 1979 and 2019 produced by box averages; years of heat wave occurrences are highlighted.





assemble a summer (JJA) mean of 2m-air temperature anomalies on land and SST anomalies over the ocean, where the long term trend (1979 - 2019) is removed, visualized in Fig. 2a. Closed boxes refer to the area used by Duchez et al. (2016), whereas dashed boxes capture the regions where 2018 anomalies were centered.

In summer 2018 the North Atlantic ocean exhibited SST seasonal mean anomalies of up to -2.5°C, whereas Europe experi-
enced 2m-air temperature seasonal mean anomalies of about +2°C. Negative SST anomalies are slightly shifted towards the Northwest compared to 2015, corresponding to a displacement of the box for largest anomalies in summer 2018. Positive surface temperature anomalies are mainly spread over central Europe during summer 2015 (Duchez et al., 2016). Instead, summer 2018 shows an extension of positive temperature anomalies towards Scandinavia, justifying a stretching of the 2018 box towards higher latitudes. However, the overall SST anomaly patterns and European temperatures in 2018 resemble those
in Duchez et al. (2016) for the summer 2015 (see their Fig. 1a), with cold anomalies in the North Atlantic and warm anomalies over Europe, although in slightly displaced regions.

Figure 2b explores whether the pattern seen in the summers 2015 and 2018 is seen in other years as well and shows time series of yearly JJA anomalies for the four different boxes from Fig 2a. Beside 2015 and 2018, the European heat wave in 2003 shows a slight anti-correlation between SST and 2m-air temperature. By contrast, the Moscow heat wave in 2010 does not reveal an
anti-correlation between anomalies of both parameters, but positive SST values of +0.6°C.

The two box averages based on anomalies in 2015 and 2018 yield only slight differences in 2m- air temperature during the heat waves 2003 and 2010. The average over the 2018 box (dashed line) produced even negative surface temperature anomalies during summer 2015, as the heat wave was constrained to central Europe. By contrast, the temperature anomalies are enhanced for summer 2018, consistent with a widespread warming that extends towards Scandinavia (Fig. 2a).
Previous studies suggest that negative anomalies of the North Atlantic SST are favorable for a phase-locked meander in the jet stream (Duchez et al., 2016; Mecking et al., 2019). Next, we investigate whether these anomalies are linked to a persistent jet stream pattern, that triggers positive European surface temperature anomalies.

## 3.2  Influence on jet stream properties

In order to evaluate a link between North Atlantic ocean anomalies and the jet stream behaviour, we perform a composite study.
Fig. 3a illustrates the behaviour of the jet stream maximum during the 25% coldest (blue) and warmest (red) North Atlantic SST months. For detailed information about the composition we refer to the section 2.2. The zero line depicts the case of the reference, the medium SST events.

Variations in the jet stream between cold and warm composites are continuous over the North Atlantic: the jet stream maximum based on cold SST events shows a steadily southward displacement west of 0° with a local minimum of about -1.5°
at around 30°W, corresponding to a preference of a trough in this region. The jet stream based on warm SST events reveals a slight northward displacement west of 0°. Further downstream (0 - 40°E) the signs of both composites are reversed with similar displacements of roughly +1° for cold and -1° for warm SST composites. Between 40 and 90°E the composites exhibit similarly shifted jet stream maxima. A reduced composite size by selecting only the 0.1 and 0.9 quantiles for cold and warm composites, respectively, yields a similar picture with an even more undulating jet stream for the cold composite (see Fig. S1 in



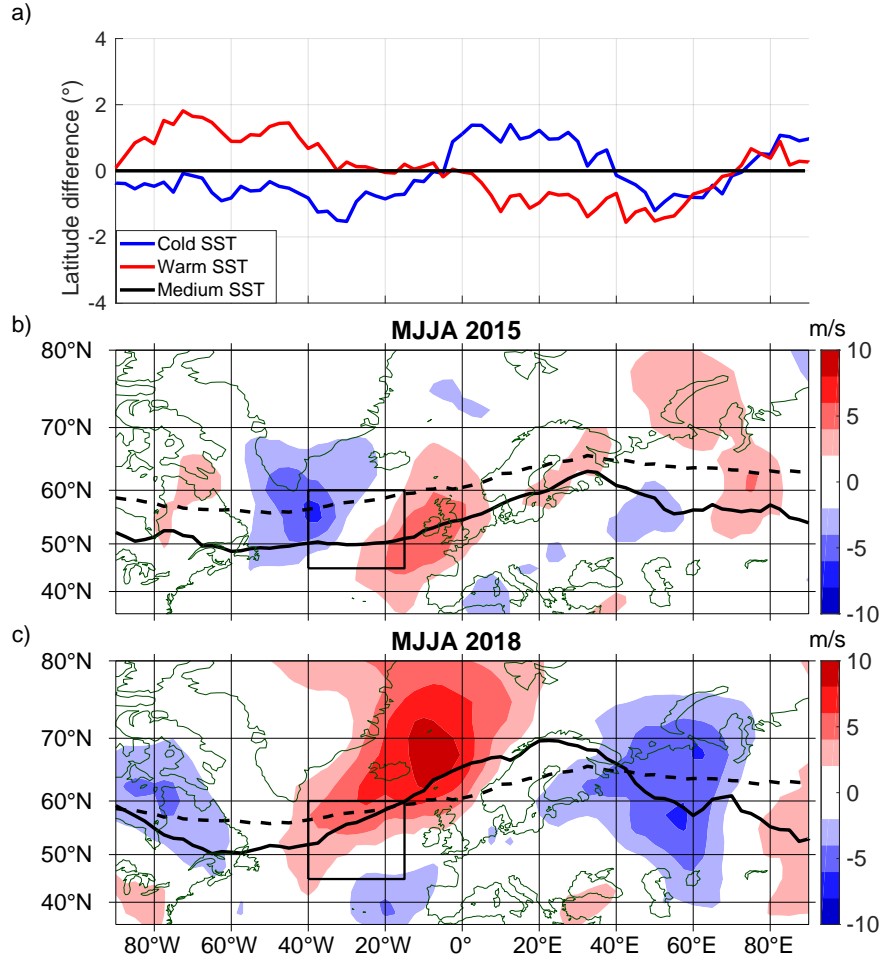

**Figure 3.** a) Composite of MJJA 300 hPa jet stream latitude (wind speed maximum between 45 and 90° N) anomalies for cold North Atlantic SST events (blue - 0.25 quantile) and warm North Atlantic SST events (red - 0.75 quantile) with respect to medium SST events (black); b) MJJA 2015 300 hPa total wind speed maximum for each longitude (solid) and total wind speed maximum for medium SST events (dashed); shading indicates 300 hPa meridional wind anomalies (MJJA 2015); c) same as b) but here for MJJA 2018.

the supplement). Particularly the southward shift of the jet stream in the North Atlantic as well as over Asia is more pronounced with values of less than -2° latitude.

In summary, cold SST events generally show a wavier jet stream and a well-developed trough-ridge system over the North Atlantic-European sector. Warm SST events show the opposite behaviour with a weaker amplitude.

During summer 2015 as well as summer 2018 cold SST anomalies were present in the North Atlantic region. Fig 3b and c

display the behaviour of the jet stream during those periods. In summer 2015, the jet stream maximum (solid line) runs south of the climatological jet stream position (medium SST composite, dashed line). Nonetheless a stronger southwest-northeast slope,





reaching from the Northeast-Atlantic until the local maximum in the northwestern tip of Russia, intensifies the trough-ridge pattern in the North Atlantic-European sector, supported by a pattern of negative and easterly adjacent positive meridional wind anomalies of 4-6 ms$^{-1}$ in this sector (coloured).

The summer 2018 reveals an even stronger picture: the jet stream maximum shows a stronger than usual trough over New-foundland at 50°N and a ridge extending up to the northern tip of Scandinavia at around 70°N. These results are in agreement with Kornhuber et al. (2019), who illustrated significant and widespread meridional wind anomaly patterns, which support the exceptionally strong waviness of the jet stream, appearing in the North Atlantic-European sector. Near-zero phase speeds and a preferred phase position of the wave, consisting of a trough over the North Atlantic and a ridge over the British Isles,

essentially contribute to the generation of the European heat wave occurrence in summer 2018.

Both, composite and case studies verified that cold North Atlantic SST patterns could be at least partly responsible for a stronger waviness of the jet stream in general and a higher likelihood of a ridge over Europe. Haarsma et al. (2015) supports that a cold anomaly in the subpolar North Atlantic ocean induces a reduction of turbulent heat release by the ocean to the atmosphere. Further it will favour a high pressure anomaly located downstream of the cold anomaly, roughly around the British Isles. Such

a pattern further resembles the positive phase of the summer North Atlantic oscillation (summer NAO) (Folland et al., 2009). Next, we examine the relationship between the North Atlantic SST, the upper troposphere and the European surface temperature with the aid of lead-lag correlations.

### 3.3 Lead-lag correlations

The verification of a link between the North Atlantic and the upper troposphere is done by using lead-lag correlations between

a North Atlantic SST box average and the 300 hPa GPH field for the whole time range (1979 - 2019). Further information about the methodologies are described in section 2.2.

The climatological correlations in the zero-lag configuration (Fig. 4a) indicate a dipole structure between the North Atlantic and Europe: positive correlations of up to 0.4 in the region of Greenland and Iceland are adjacent to slightly negative correlations over Europe with values of roughly 0.3. These findings confirm that an initial cold SST anomaly spreads across the North

Atlantic is likely to coexist with an upper-tropospheric trough over the North Atlantic and accompanied by a ridge downstream over Europe. The dipole pattern of climatological correlations maintains the position and strengthens with positive correlations of up to 0.5 when the SSTs lead the circulation (Fig. 4b-d).

Hall et al. (2017) confirm that low frequency variability associated with the North Atlantic SST leads the summer Atlantic atmospheric state, including the location and strength of the jet stream. The study by Ossó et al. (2018) supports that the

atmospheric response to the SST has an equivalent barotropic vertical structure by evaluating the geopotential height anomalies at different pressure levels during summer.

We also discuss correlations between the North Atlantic SST and the 2m-air temperature, displayed in Fig. 5. The European box denotes a box used for an average and subsequent lag analysis. Climatological correlations reveal again a dipole structure in the North Atlantic-European sector. Strong positive correlations of +0.8 are present around the North Atlantic box. Values are

approaching zero with greater distance from the box. Central and eastern Europe is covered with slightly negative correlations





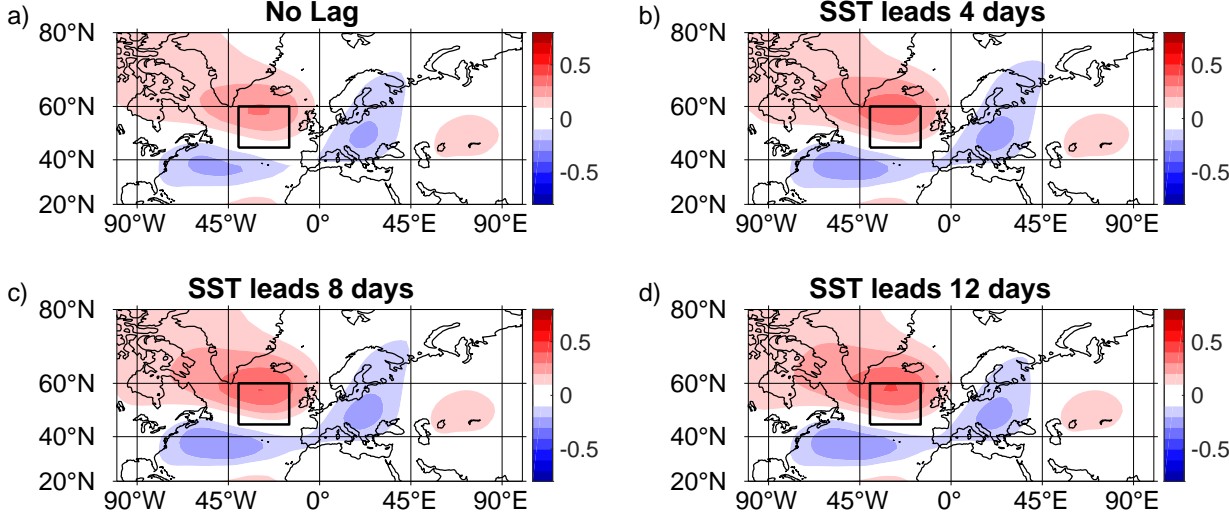

**Figure 4.** Correlation maps between daily values of SST anomaly of a North Atlantic box (15 - 40°W and 45 - 60°N) and 300 hPa geopotential height for the whole time range (1979 - 2019); both parameters are detrended, deseasonalized and smoothed with a 15-day mean: figures show zero-lag and leading SST (4,8 and 12 days) configurations.

of approximately 0.2. The values increase again, when the SST leads the circulation (Fig. 5b,c). 12 day-leading configuration shows a slight decline in the correlation (Fig. 5d). Correlation analysis of the North Atlantic SST and GPH (Fig. 4) and 2m-air temperature (Fig. 5) produced patterns with almost similar locations.

The study by Haarsma et al. (2015) further confirmed that an initial ocean induced pressure response due to a reduced ocean

atmosphere heat release in the eastern part of the North Atlantic contributes to summer circulation changes over western Europe, consisting of reduced rainfall and cloud cover as well as a strengthened incoming solar radiation and thus enhanced surface temperatures. Duchez et al. (2016) argues that not the cold North Atlantic anomaly itself, but the resulting enhanced meridional SST gradient favored the high pressure dominance and temperature extremes over Europe. Another approach was made by the study of Dunstone et al. (2019), who extended the European high-pressure response even to a tripole pattern of

North Atlantic SST's.

In order to understand the consecutiveness between North Atlantic SST and GPH as well as the European 2m-air temperature, we attribute particular emphasis on the quantitative description of the lagged relationship. Fig. 6 illustrates the correlation between the parameters with respect to the lag. Averages of correlations between SST and 300 hPa GPH are made based on the Atlantic box and between SST and 2m-air temperature based on the European box. Climatological correlations used for

the previous figures 4 and 5, are averaged and indicated by the black lines. In agreement with the previous findings from Fig. 4, we obtain a positive correlation between the SST and the 300 hPa GPH within the same North Atlantic box. In coincidence





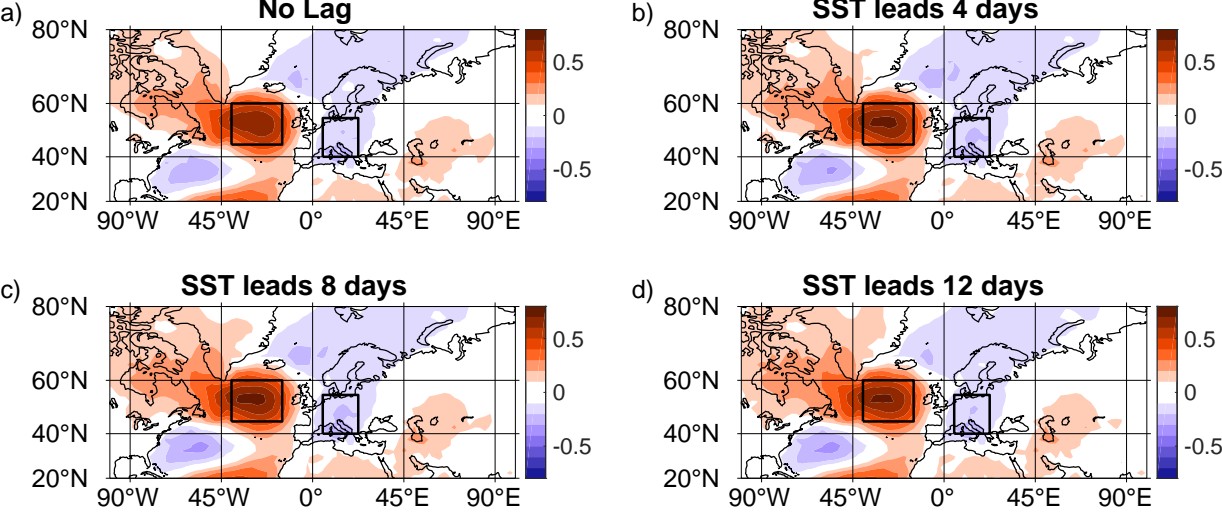

**Figure 5.** Same as for Fig. 4, but here for correlations between SST anomaly and 2m-air temperature; a box over Europe (5 - 22.5°E, 40 - 55°N) indicates the area used for averages shown in Fig. 6.

with Fig. 5 negative correlations are present between the North Atlantic SST and the European 2m-air temperature.

Although the correlations exhibit maxima with low values of 0.25 and -0.19 for the correlation of the North Atlantic SST with 300 hPa GPH and the European 2m-air temperature, respectively, the peak correlation confirms the tendency of the circulation

leaded by the North Atlantic SST.

In addition to the climatological correlations, we include correlations for cold SST events and warm SST events, as already described for Fig. 3. Again SSTs lead the circulation, and we obtain a stronger correlation for cold SST events compared to climatology and warm SST events. A stronger correlation between the parameters during cold SST events is in line with the more amplified North Atlantic jet stream latitude anomalies present in Fig. 3 for the cold SST composite. Values of the warm

composite are generally lower or equal compared to the climatological correlations.

An altered composite size taking the 0.1 and 0.9 quantiles shows a similar picture for correlations between SST and GPH (Fig. S2). Values are again higher in the presence of cold SST events. Correlations with the European temperature do not confirm that the SST leads the atmospheric circulation, which is probably due to a not large enough sample size in this case.

Nevertheless, cold SSTs in the subpolar North Atlantic tend to be accompanied by high temperatures over Europe, as the low

subpolar SSTs are favouring a vertically barotropic (Ossó et al., 2018) pressure distribution that channels warm air northwards into Europe triggering heat wave events (Caesar et al., 2018). Subsequently we study whether the pattern evolves via a certain wave pattern related to wave resonance and amplification associated with the QRA mechanism.



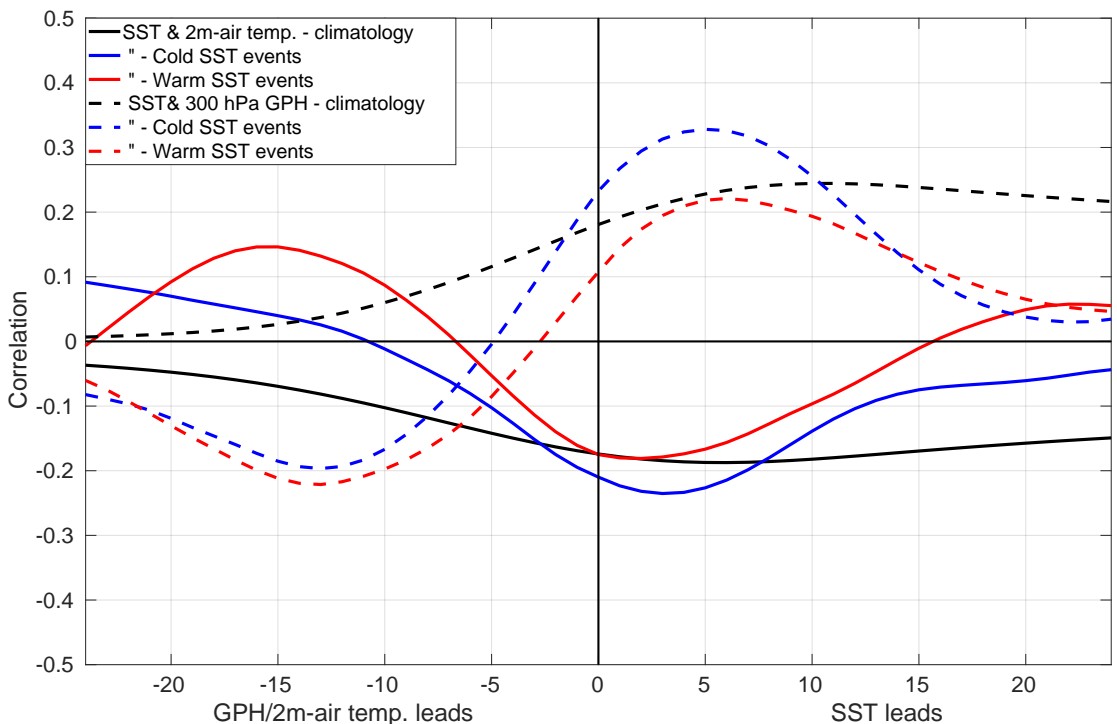

**Figure 6.** Correlations as a function of lag between North Atlantic SST box average (15 - 40°W and 45 - 60°N) and (i) 300 hPa geopotential height average for the same box (dashed); (ii) 2m-air temperature based on European box (5 - 22.5°E, 40 - 55°N) (solid); black lines represent the whole range (1979 - 2019), whereas blue and red lines indicate correlations with respect to cold (0.25 quantile) and warm (0.75 quantile) SST composites, respectively.

### 3.4 Influence on wave stationarity and resonance preconditioning

The anti-correlation between the subpolar North Atlantic SST and the European temperature combined with the upper-
tropospheric trough-ridge pattern is evident throughout the previous analyses. Now we clarify whether these surface conditions are a result of the wave resonance and amplification associated with the QRA-mechanism. A pronounced double jet structure is the basis for the onset of the process of wave resonance. Kornhuber et al. (2017) claim that the majority of detected QRA events are related to double jet structures. The North Atlantic SST anomalies occur at the same time with the double jet index (DJI) defined in section 2.1 and shown in the scatter plot in Fig. 7. The analysis is restricted to the summer season (MJJA).
Monthly values of recent summers with European heat waves (2003, 2010, 2015, 2018) are highlighted within the plot.
The ocean's property of low frequency variability is responsible for the persistence of SST values within a certain range throughout one individual summer. For instance, the summer SST anomalies in 2018 are constrained between -1.1 to -0.5°C.



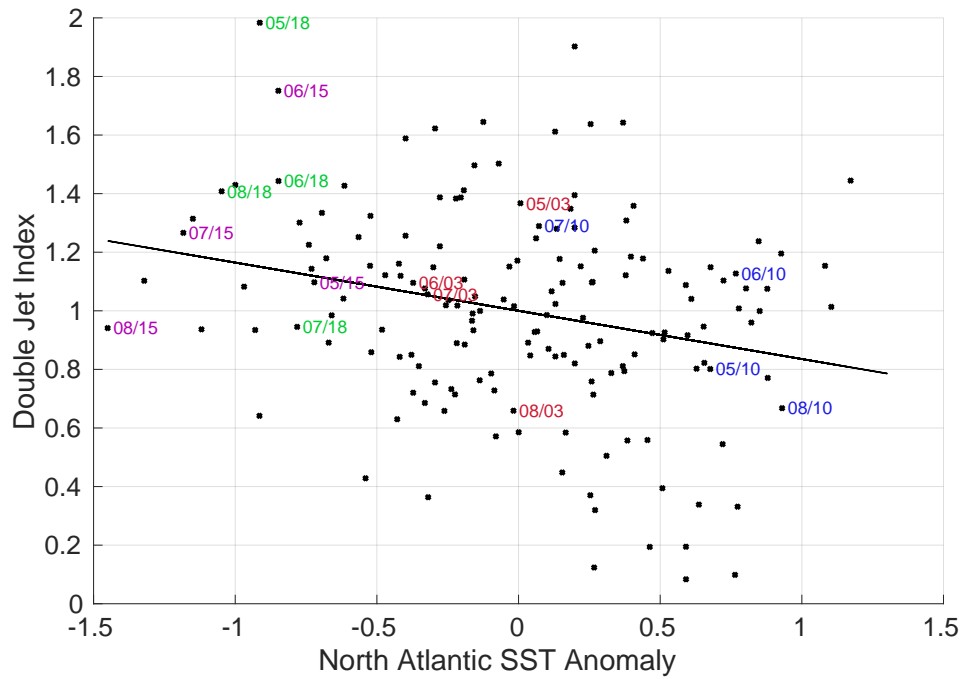

**Figure 7.** Scatter Plot of monthly mean values (MJJA) for the DJI and North Atlantic SST anomaly (15 - 40°W and 45 - 60°N); years of particular interest are denoted by 'month/year'; the black line represents linear regression.

Despite a strong scattering, the tendency of a more intense double jet pattern during cold SST anomalies is evident, supported by the tilted line of linear regression with a slope of -0.22. As an example serves the case study of 2018, where a negative North
Atlantic SST anomaly shows the higher possibility for a stronger double jet than usual. Particularly May 2018 stands out with a double jet pattern, that is twice as strong as the climatology.

Once a double jet pattern is established, next stages of quasi-resonant amplification can develop (Kornhuber et al., 2017; Coumou et al., 2018). Advanced analysis of the wave pattern is done by inspecting the wave amplitude and phase speed. Detailed information about the derivation is provided in section 2.3.

The JJA climatology (black contour lines) features the preference for eastward-traveling waves (positive phase speeds) (Fig. 8). Quasi-stationary waves with a tendency of higher amplitudes as well as a fraction of westward-traveling waves are further evident from the spectrum, which is in agreement with the study by Kornhuber et al. (2017).

Assuming that QRA events are present, we expect positive PDF anomalies in the area of low phase speeds and high amplitudes and negative anomalies of low-amplitude fast waves. The study by Kornhuber et al. (2017) found these conditions in the spectra
of wave numbers 6 to 8 and Kornhuber et al. (2019) explicitly show wave-7 activity during the European heat wave of 2018. By compositing low SST and high DJI values as a potential precursor for QRA events, either separately or together, we found



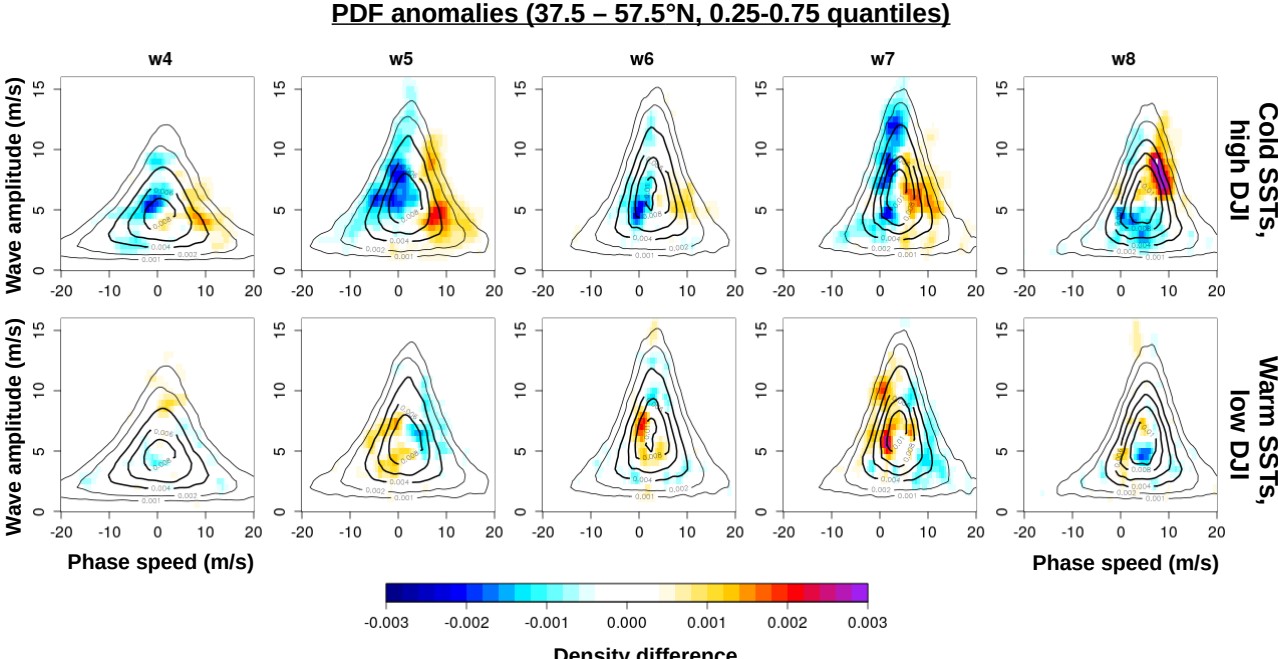

**Figure 8.** Probability density function of phase speed versus wave amplitude (derivation described in section 2.3); anomalies are shown for the 10% coldest SST days in the North Atlantic box (15 - 40°W and 45 - 60°N) together with the 10% highest DJI values selected; wave properties are shown for 37.5 - 57.5°N for comparison with Kornhuber et al. (2017); black contour lines denote the JJA climatology.

no consistent anomaly in the context of the QRA theory. Instead, we show that cold SST events combined with high DJI values enhance the dominance of eastward-traveling waves and reduce the occurrence of quasi-stationary high-amplitude waves (Fig. 8, upper row), apparent throughout all PDFs and particularly for wave number 5,7 and 8. Thus, Fig. 8 does not find any QRA

imprint, as the composite of cold SSTs and high DJI values does not show higher occurrence of quasi-stationary and amplified waves. This is an unexpected result as high DJI is supposed to trigger QRA events.

Composites based on warm SSTs and low DJIs (Fig. 8, bottom row) exhibit the reversed signature, but with a lower magnitude. Based on these results, the jet stream latitude anomalies shown in Fig. 3 consistent with the correlation patterns from Fig 4 to 6 have to be explained by a different mechanism.

In order to probe the behaviour of transient waves preferred during cold North Atlantic SSTs, we take the 10% coldest SST events and calculate the difference between troughs and ridges appearing over the North Atlantic shown in Fig. 9 (coloured). The anomalies are in general higher with respect to the previous ones shown in Fig. 8. Here, positive anomalies concentrate at near-zero phase speeds throughout all wave numbers, implying that a trough present in the North Atlantic slows down the preferred eastward-traveling waves during cold SST events. This would result in a longer duration and a higher occurrence of

a trough over the North Atlantic box during cold SST events. When classifying the here identified slow-moving waves during

**Trough – ridge PDF difference over cold N. Atlantic SSTs (37.5 - 57.5°N)**

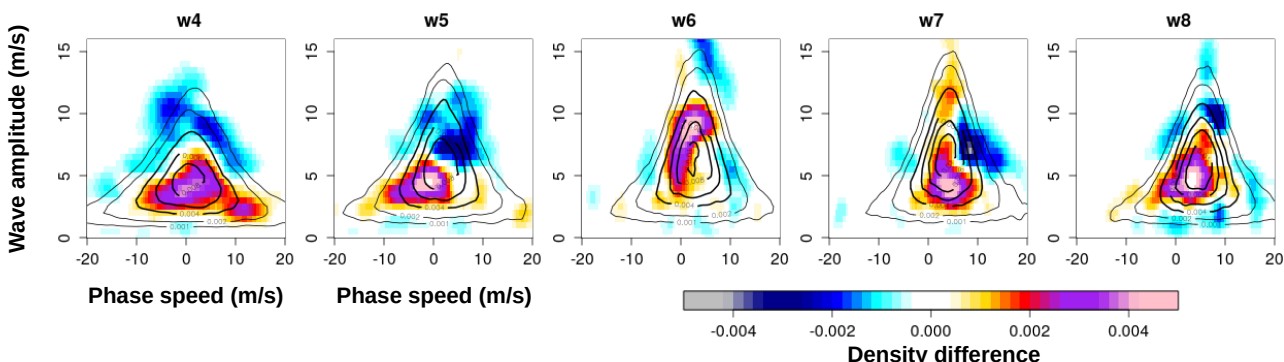

**Figure 9.** Same as in Fig. 8 but here the 10% coldest SST days in the North Atlantic box (15 - 40°W and 45 - 60°N) are selected and then separated whether trough or ridge are present over this box; anomalies (coloured) illustrate the difference between trough and ridge.

a trough and cold SST present in the North Atlantic into the context of the QRA theory, it would require positive anomalies at higher amplitudes. But only the wave-6 spectrum indicates a slight imprint of high-amplitude waves. The remaining wave patterns even show an enhancement of low amplitude waves, which is contradictory to the patterns found in the study by Kornhuber et al. (2017). While they studied the PDFs of wave numbers 6 to 8 for QRA events, we can provide a consistent

pattern by inspecting even wave numbers 4 to 8 including medium-scale waves. A similar result is obtained using a different latitudinal section of 50 -70°N, shown in Fig. S3. Note that anomalies shown Fig. 9 and S3 illustrate the difference between troughs and ridges, whereas the overall appearance combining them is provided in Fig. 8.

The relationship is not linear, as the counterpart, the 10% warmest SST events, does not show the opposite signature in trough-ridge differences (see Fig. S4). Additionally the anomalies are less pronounced during the warm SST composite. We investigate

the wave amplitude in the North Atlantic box relative to the zonal mean by a decomposition into different quantiles (Fig. 10). Probabilities of a trough (negative values) or a ridge (positive values) present in the North Atlantic are determined using the 300 hPa GPH.

The JJA climatology is characterized by a nearly symmetric distribution and a maximum, which is in the area of negative amplitude values, implying that the North Atlantic box has in general a very slight preference for a trough. Colder SST composites

(the lowest quantiles) show a tendency to shifting the whole probability distribution towards negative values, i.e. deeper and more long-lasting troughs. The opposite happens with the warmer SST composites (higher quantiles) with a distribution shift towards positive values, but to a lesser extent.

These results are consistent with the correlation figures (Fig. 4 and 5): a trough dominant during cold events and a ridge present during warm events supports the positive correlations between the SST anomaly and the 300hPa GPH within the North Atlantic

box.

In this study, we were looking to confirm the relationship between the QRA mechanism and North Atlantic SSTs. We found



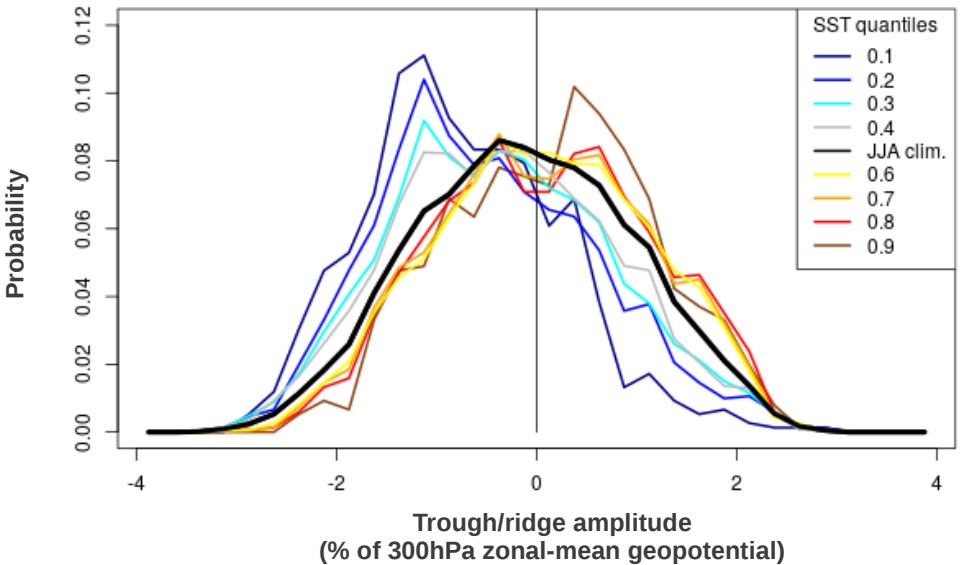

**Figure 10.** Probability of daily North Atlantic trough and ridge amplitude represented as the anomaly of 300 hPa geopotential height over the North Atlantic box (15 - 40°W and 45 - 60°N) relative to the zonal mean of the same latitudes; an exact description is provided in section 2.3; values are decomposed into the different quantiles based on the calculation described in section 2.2.

that high DJI values (Fig. 7) would provide a necessary precondition for the development of wave resonance and amplification. Favorite conditions of the QRA theory are evident in the jet stream latitude composites as well as in the correlation plots (Figs. 3 to 6). But no QRA signature appears in the phase speed versus amplitude PDFs shown in Fig. 8 in contrast to the study

by Kornhuber et al. (2017). We do not find higher amplitudes and stationary waves in cold SST and/or DJI composites. We explain the favorite conditions for QRA found in Figs. 3 to 6 by transient waves being slower (but not necessarily stationary or amplified) while a trough is present over North Atlantic cold SSTs, apparent from Figs. 9 and 10. The higher amount of time spent with this setting leaves its signature on the jet stream latitude composites (Fig. 3) and correlation plots (Fig. 4 to 6).

A given DJI preconditioning in individual events may be a subject to the QRA mechanism. But in our study the QRA mech-

anism, that is meant to be related to wave resonance and subsequent amplification is not found to be responsible for the jet stream latitude anomalies and European temperature correlations related to anomalously cold North Atlantic SSTs. Our findings include the slow down of waves, while a trough and cold SSTs are present in the North Atlantic, which might be essential for inducing European heat waves, but does not involve an imprint of the QRA theory.

## 4 Conclusions

An increase in European heat waves during the 21$^{st}$ century gained a special attention about the drivers of these anomalous conditions. The study by Duchez et al. (2016) proposed a relationship between significantly cold SST anomalies, found in the





eastern North Atlantic, and the European surface temperature via Rossby wave propagation. Another study by Kornhuber et al. (2017) suggested that heat waves are referred to the QRA mechanism associated with wave resonance and amplification. Our study combined the theories by Duchez et al. (2016) and Kornhuber et al. (2017) for the first time. We summarize our main

findings as following:

1. European heat waves, particularly the occurrences during 2015 and 2018 both reveal a widespread area of negative SST anomalies across the eastern North Atlantic and positive surface temperature anomalies in central to northern Europe in agreement with Duchez et al. (2016).

2. A composite study as well as the case studies of 2015 and 2018 provide evidence of an enhanced undulation of the jet
stream during cold North Atlantic SST events, phase-locking the North-Atlantic European sector into a trough-ridge pattern.

3. Correlation analysis reveals that the North Atlantic SSTs correlate on the one hand with the upper geopotential height signal within the North Atlantic and anti-correlate on the other hand with the European surface temperature. Further we found that the SST leads the summer atmospheric circulation for about a couple of days to one week (Fig. 6).

4. Cold North Atlantic SST anomalies promote increased double jet occurrence, a necessary precondition for wave resonance associated with the QRA mechanism. However our wave analyses with phase speed versus amplitude PDF's do not find any QRA imprint related to cold North Atlantic SSTs, compositing cold SSTs, high DJIs or both together.

5. We find that cold North Atlantic SST events decelerate transient waves, which are preferred during those events. Slower but still traveling waves (and not necessarily higher amplitude) are responsible for the climatological imprint (Figs. 3a
and 4 to 6) when cold SST's are present with an upper-level trough, provoking a more persistent ridge downstream over Europe and initiating long-lasting heat conditions subsequently.

6. The relationship is not linear, as the warm SST events do not show a reversed signal in terms of phase speed. Regarding the preference of a ridge in the North Atlantic during warm events, the magnitude is less pronounced compared to the case of cold SST events.

The contribution of the SST to variations in the atmospheric state are not only important in the year-to-year variability, but also on a longer time scale, since global warming will increasingly leave its mark: understanding the atmospheric effect of the North Atlantic warming hole (Drijfhout et al., 2012) as a long-term negative SST anomaly seems to be essential for the summertime circulation and the potential development of European heat waves.

*Author contributions.* JK performed the double jet index (DJI), composite and correlation analysis, produced Figs. 1 to 7 and wrote the
manuscript; RPK assisted in the methodology development, designed the wave analysis, produced Figs. 8 to 10 and commented on the manuscipt; KM motivated the study, supervised its development and commented on the manuscript; KB contributed with ideas and suggestions.



*Competing interests.* The authors declare that they have no conflict of interest.

*Acknowledgements.* We thank the European Centre for Medium-Range Weather Forecasts (ECMWF) data server for the freely available
ERA-5 reanalysis data.




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
