# Peer review of "Impact of North Atlantic SST and Jet Stream anomalies on European Heat Waves"

_Weather and Climate Dynamics, 2020_

## Referee Comment (RC1) · Anonymous Referee #1 · 21 Aug 2020

The current work considers anomalies of SSTs and the upper tropospheric waviness in the North Atlantic sector in an attempt to estimate their impact on the occurrence of European heat waves. Specific aspects of both factors have been suggested in the past to be conducive to European heat waves, but this paper considers, for the first time, both factors simultaneously. A novelty hereby is the fact that the authors focus on the waviness over the North Atlantic rather than the circumglobal waviness as was common in numerous previous studies.

The manuscript contains a number of interesting aspects, like the lag-correlation analysis in order to tease out a cause-and-effect relationship or the discrepancies with respect to previous work regarding the wave behavior in connection with European heatwaves. There seems to be potential for progress in these directions. At the same

time, I see a number of fundamental weaknesses. First of all, the local double jet index introduced by the authors seems to be incompatible with the idea of quasi-resonance and, therefore, not an adequate metric to quantify the latter. Also, the analysis is mostly based on composites and correlations. A correlation does not imply causation, and therefore the analysis does not live up to the expectation raised by the title ("Impact of .... on ...."), which promises to shed light on causal connections. In addition, many of the shown correlations are pretty weak; in no case do the authors investigate whether the correlations and the patterns on the composite maps are statistically significant or not. Throughout the text the authors relate their results to those of earlier studies, and they sometimes find agreement and sometimes disagreement. In the end this left me lost and I could not find out what I am supposed to learn: are some of the earlier results wrong (because of the detected disagreement), or are the authors' own results an outcome of an inadequate metric (see above) or pure chance (which would be consistent with a potentially missing statistical significance) or both?

Below I provide further details under "major issues" in order to illuminate my concerns. Items 3–5 could be addressed by major revisions. Item 2 seems critical, although one might argue that a subsequent proof of statistical significance (if that turned out to be possible) would save the day. However, item 1 is fundamental in the sense that a substantial part of the analysis is based on the local double jet index, which in my eyes is inappropriate to fulfill its purpose. Addressing all these issues will basically result in a very different manuscript. I thus have to recommend rejecting this manuscript.

I chose not to mention further minor issues.

Major issues

1. The theroy of "quasi-resonance" requires Rossby waves to travel around the entire globe. This is a fundamental aspect of quasi-resonance, because the singularity in the case of no damping (see equation (3) in Petoukhov *et al.* 2013)

results from the constructive interference of an infinite number of waves, which is only possible if the domain is periodic in longitude and if the waves see a zonal waveguide at all longitudes. Correspondingly, previous authors have analyzed the zonally averaged zonal wind and computed the corresponding stationary wavenumber $K_s(\phi)$, which depends on latitude $\phi$. According to WKB theory, a relative maximum in $K_s(\phi)$ — which typically occurs in the presence of a double jet structure — is then indicative of a zonal waveguide all the way around the globe.

In the current manuscript, the authors diagnose the double jet structure locally over the North Atlantic ocean. This leaves open two issues: first, does this also correspond to a relative maximum in $K_s(\phi)$? And, second (more fundamentally), even if there were a local (in longitude) relative maximum in $K_s(\phi)$, this alone would not indicate a wave guide all around the globe. Therefore, it seems to me that the authors' local double jet index is inadequate for diagnosing a zonal waveguide as required for the validity of quasi-resonance theory.

2. One key concern is that many of the shown effects (correlations, composites) are rather weak and it is not clear to me whether they are statistically significant of not. It appears necessary to me to determine the statistical significance of all these results before they can be presented and discussed in a scientific paper. For instance: Are the differences shown in Fig. 3 statistically significant? Are the weak correlations shown in Figs. 4 and 5 statistically significant? The same for the correlations shown in Fig. 6, which are even smaller (a correlation coefficient of the order of $r = 0.2$ corresponds to a "described variance" of $r^2 = 0.04 = 4\%$, which is very low and requires a large and high-quality data set to tease it out in a significant manner).

The issue of statistical significance and the interpretation of a correlation in terms of causality can be illustrated in Figure 2. Apparently, some heatwaves over Europe are associated with cold SSTs, some are not, and sometimes there is

a cold SST anomaly but no heat wave. Overall this is a very mixed bag, and I believe that the correlation between two respective time series is very low and probably not significant. On the other hand, if cold North Atlantic SSTs were really "drivers" ("impact", "influence", etc.) of European heatwaves, wouldn't this correlation have to look more convincing?

3. The authors talk about "impact of ... on ...." (title), about "drivers of ...." (second paragraph in the introduction), about the "influence on ...." (section 3.1 heading), "A promotes B...." (conclusion section) etc. All these terms suggest that this paper is about identifying causes for European heat waves. Correspondingly I started to read the paper with high expectations and was disappointed later on, because in my eyes the kind of analysis presented in the manuscript does not allow one to determine the causes for heat wave occurrence. As I said above, correlation does not imply causation. More specifically, I do not believe that the "influence of ... on ..." can be "evaluated by using a composite study" (as promised, e.g., on line 96). An analysis based on composites or on correlations is at best suggestive, but certainly not conclusive. To give an example: the plot shown in Fig. 3 indicates an association between North Atlantic SST and the upper tropospheric flow over that region in two particular years. The questions that remains unanswered: are these two years just a fortuitous coincidence (see my earlier issue about statistical significance), does the ocean "drive" the atmosphere, or does the atmosphere "drive" the ocean? In summary: given what the authors show in the manuscript, I think that the language is inflated and the analysis does not live up to the expectations raised by some of the formulations.

4. Some of the reasoning seems rather casual. For instance, the authors extend the analysis of a previous study to the year 2018 and show that the European heat wave was associated with a simultaneous cold SST anomaly over the North Atlantic similar as in that previous study for a previous year. So far so good. On the other hand, Fig. 2 shows that some other extreme heat waves such as the

2003 or the 2010 heat wave were *not* associated with strong cold SST anomalies over the North Atlantic. So what am I supposed to learn from that? Does it mean that the association between SST anomalies and European temperatures is a random event? The authors do not provide any guidance to the reader as to how to interpret these results.

5. I have an issue with the lag correlation analysis (Figs. 4, 5, and 6) in combination with preprocessing the involved variables through a 15-day running mean. A running mean is a *non-causal filter* which projects information backwards in time. I do not say that the results shown in Fig. 6 are invalid, I am just skeptical and it would be more convincing if the same results could be obtained without a non-causal filter. Also, the four plots in Figs. 4 and 5, respectively, are very similar to each other, and it appears to me that this is in essence a result of the 15-day running mean filter preprocessing: the latter smears out the data in the time domain such that a time lag of 4 or 8 (or even 12) days becomes almost invisible.

In the end the issue goes back to the question of statistical significance (see one of my previous items). For instance, the effective number of degrees of freedom in Fig. 6 is likely to be very low as a result of the 15-day running mean. This makes me believe that a difference in the correlation coefficient of as little as 0.2 between a 15 day lead and a 15 day lag could be statistically insignificant.

**References**

Petoukhov, V., S. Rahmstorf, S. Petri, and H.-J. Schellnhuber 2013. Quasiresonant amplification of planetary waves and recent Northern Hemisphere weather extremes. *Proceedings of the National Academy of Sciences 110*(14), 5336–5341, doi:10.1073/pnas.1222000110.

---

## Referee Comment (RC2) · Anonymous Referee #2 · 2 Sep 2020

**1 General comments**

The submitted manuscript covers important dynamical aspects on the evolution of heat waves and tries to link those to understand the underlying mechanisms. The authors focus on previously identified potentially important drivers of heat waves, namely the wave amplification according to the quasi-resonance mechanism and Atlantic precursor SST anomalies. Those feature have been previously shown to be linked to some heat waves. The authors further investigate the role of jet stream anomalies as link between those features and try to answer the question about their importance on heat waves. The study nicely combines the investigation of two case studies (heat waves in 2015 and 2018) and a more general climatological perspective of a 40 years period using ERA5.

The study is nicely motivated and it picks up two (three) mechanisms which are further investigated in detail and the authors tried to investigate a possible link between them. This setup of the paper appears quite exciting, however, the manuscript still needs some substantial improvement to more convincingly support the claims they present with their results. One major issue I have with the study is that their quick conclusions about causal relationships is not made very convincing. I don't think that the presented results are enough to support their causal relationships (specific comments about this further below). This seems to be even more problematic that the results are not supported by any means of statistical significance test. As mentioned further below for the specific comments I did struggle to understand if the shown patterns are really meaningful enough to come to the conclusions by the authors. I think applying some measure of statistical significance could be helpful for the authors to make their point more convincingly and convince the reader that their results are indeed meaningful. A related issue I have with this is the missing understanding of sensitivity of the results to some of their choices. I made this point also in the specific comments, like for example for the choice of double jet index. I would suggest that the authors should either refer to another study where this index has been used, or at least motivate their choices or tell the reader about the sensitivity of the results on those subjective choices. This issue appears several time during the manuscript and is also enhanced by the authors themselves, by showing that changing the averages boxes for SST or temperature can lead to different results. This raises the question, how much the following results can be trusted or if a meaningful interpretation is difficult due to this sensitivity. I would suggest that the authors give the reader further insight into the impact of these changes.

Further, I did struggle a bit to understand what the actual real goal of this paper is, related to wave amplification mechanisms. The quasi-resonance mechanism is a possible mechanism relevant for heat waves, but it seems that the authors are interested in the dynamics between different contributors in the Atlantic to support the evolution of heat waves, SST anomalies, jet structure and evolving wave patterns. This impression is further strengthened by the authors choices to deviate from the procedure of the quasi-resonance mechanism on a more locally restricted scale (in terms of extension of double jet structure). But why then not using a more general wave measure? Previous

literature has shown that not all heat waves are associated with global scale waves, but more locally restricted ones. I assume the reason is that the authors were only interested in this sub-class of waves associated with heat waves, which is fine, but I think that the authors then should make this point more clearly and highlight that they specifically are only focusing on such waves. This includes also to clarify why the authors nevertheless deviate in their search for dynamical links to locally restricted jet anomalies. So the authors should clarify their choices, that they investigate potentially important aspects that can lead to heat waves, but also why they link those without clarifying why the authors think it is reasonable to link and consider explicitly those features together?

In general, I think the manuscript represents a nice study which tries to understand specific aspect and their dynamical links that can lead to European heat waves. However, at the moment the conclusions of the paper are not presented convincingly. This could be improved by better explain to the reader the motivation behind some of their diagnostics, including some support from including some kind of statistical significance test, comments about the sensitivity of the results and being a bit clearer on the aim of this paper (related to the applied diagnostics).

**2 Specific comments**

**2.1 Section 1 (Introduction)**

**lines 60-62:**
The aim of this paper is not fully clear to me and I am slightly confused about the focus between SST anomalies, heat waves and QRA. These two sentences are a good summary for this slight confusion, because it seems like the authors have some underlying assumptions that are not explicitly made clear to the reader. According to the second sentence (lines 61-62), the research objective is to investigate the connection between European heat waves and North Atlantic SST anomalies with the double jet structure as possible dynamical link between those two (with the associated wave properties that can be linked to the jet structure). Therefore the authors explicitly want to investigate the link between Duchez et al. (2016) and Kornhuber et al. (2017). I find it difficult to understand this reasoning. If I understand the mentioned studies correctly, Duchez et al. (2016) try to link the SST anomaly to heat waves, resulting from some form of atmospheric wave signal that can be linked to the SST anomaly. The QRA proposes a specific subset of waves which are assumed to be important for the occurrence of heat waves. However, the QRA is a subset of waves and there are also more general studies linking atmospheric waves to European heat waves. As the authors are interested in linking SST, jet structure, waves and European heat waves it is not clear why the study attempts to use the QRA as the underlying wave mechanism of relevance. E.g. in Wolf et al. (2018) the authors showed on the one hand a possible relevant contribution from the QRA to extreme positive temperature anomalies over Europe during NH summer, but also mentioned on the other hand that a specific contribution of the waves connected with extreme events were a rather local feature which does not so well fall into the QRA

perspective. As the study of Duchez et al. (2016) is not depending on the QRA nor did they make this connection, it is not clear why the authors focus explicitly on this link. So the main objective is to understand the QRA and trying to identify possible underlying processes for this QRA, for which the authors focus on the SST anomalies highlighted by Duchez et al. (2016)?

**2.2 Section 2 (Data and Methods)**

**lines 66-69:**
The objective of the study is to understand the dynamical link between SST, jet structure and temperature anomalies. Wouldn't it therefore make sense to subtract the climatological trend from the underlying variables to not run into danger to get a bias into the results for more extreme warm events occurring later in the investigated periodic?

**lines 74-76:**
As previously mentioned, I find it difficult to understand on the one hand the focus on the QRA, but on the other hand deviating from it to investigate rather more local processes. This somehow does a bit contradict with explicitly focusing on the global QRA feature. The authors should probably be clearer on what their study is aiming for - what is it exactly they try to understand.

**lines 82-83:**
In the previous paragraph the authors explained how they identify the three climatological branches of the jet structure for May/June and July/August from the climatological state. So the branches define these three step functions (black lines with dots) in Fig. 1, correct? In lines 82-83 the authors explain how they calculate the branches for each month, but I thought the branches are derived from the climatological fields. Do the authors intended to explain how they calculate the associated U-values for each month based on the climatological branches? Maybe the authors could make this description a bit clearer.

**line 94:**
Is there a reasoning behind the specific choice of DJI or do the specific not really matter? It is some kind of curvature (Unorth - 2*Umid + Usouth), but with more weight to the absoulte maxima of the jet maxima. Would the use of the curvature instead have an impact on the results?

**line 97:**
How do the authors remove the seasonal variability? Do they use a daily based climatology? I think the authors should specifically mention this.

**lines 104-105:**
What does this exactly mean (daily-based detection of the jet stream)? Is the maximum calculated from the composite fields and daily based refers to applied timelags for this

composite, or is the location of the maximum represented by the average of the locations for each maximum at every day used for the composite?

**line 110 (but also valid for the other figures):**
I would suggest to order the figures in the way they are refered to in the text.

**line 110:**
Is there a reason to apply a 15 day running mean to one variable, but not the other? Why is this running mean necessary. Using the 0.25/0.75 quantiles (or even 0.1/0.9) should include enough cases to get a smooth line for the time-lagged correlations.

**line 125:**
Maybe the authors could spend a bit more time on explaining this index. What does this measure of a trough or ridge, relative to the zonal mean tells the reader. An overall southward or northwards shifted jet in the Pacific would by this measure have an impact on zanom in the Atlantic. This could result in an increase or decrease of the relative strength of the Atlantic ridge or trough, even for identical conditions in the Atlantic. How can this interpreted or what is it aimed for to explain? Also in the following the definition of a trough/ridge seems a bit surprising. Isn't this region in a climatological sense represented by higher z-values, which would mean that by this definition one could identify ridges (zanom¿0.5) for cases when a gridpoint based climatological anomaly would indicate a negative anomaly (trough).

**2.3 Section 3 (Impact of North Atlantic SST anomalies)**

**lines 155-159:**
What is the conclusion here? Must the hypothesis be rejected that cold Atlantic are leading/associated with European heat waves? Or is it necessary to analyze the spatial pattern shift in T2m and SST anomalies? The authors should be clear on their interpretation of their results. This missing interpretation also raises further questions in the following paragraph. The authors indicate that there is no clear connection between Atlantic SST and European T2m, but now they investigate if the SST is responsible for the phase-locking of the jet leading to summer extremes in Europe, which they said are not linked. I therefore suggest that the authors include some further description and interpretation of their results to avoid this confusion.

**lines 191-192:**
How do the authors come to this conclusion? Jet stream position/wind and SST anomalies are maybe not surprisingly associated, therefore a change in the jet stream position will be also associated with SST anomalies, or those SST anomalies could also be driven by a shift in the jet stream position. The authors have not investigated the causal relationship and therefore should rephrase their statement that it does not lead to a misleading interpretation of a causal relationship. The author should be clear on this and rather explicitly refer to their following lead-lag-correlations to gain further insight

into this connection.

**lines 202-207:**
These findings (Fig. 4) are a bit surprising, maybe the authors can give some further insight and interpretation of these findings. With synoptic systems usually dominating the midlatitudes (and representing the GPH fields), I would not necessarily expect such a persistent correlation pattern. As the correlation maps look nearly constant, I would also expect the same correlations for 20 to 30 days. This however would make me doubt this is a good measure to investigate the prediction of some dynamical features as the build up of a wave pattern. And does the correlation goes towards zero for negative lags? I have the same issue also with Fig. 5. The patterns suggest some longterm connections, like large scale NAO patterns which can be associated with large scale flow and temperature anomalies. But to investigate such features, a 4 day lag seems a bit strange, which would rather suggest the authors try to investigate the quick atmospheric reply to the cold SST anomalies in amplifying the wave pattern downstream. I think the authors should better describe what it is exactly they are trying to investigate and with it explain their choice of time lags. Do those kind of plots show that the SST anomalies are extremely persistent and define the overall large scale flow pattern for Europe for the whole summer. If so, shouldn't the authors also show negative lags to support their statement of a causal relationship? I could also imagine a wave pattern building up the SST anomalies, with the wave pattern amplifying itself further, which could also lead to the shown correlations, but which would also have the same correlation pattern for negative lags.

**lines 241-243:**
I think more discussion is necessary and discrepancies cannot be disregarded due to a small sample size. First, if the mechanism is relavant, it should also be visible in the more extreme cases - I thought this is the whole point, presenting a mechanism for extreme summer events. Second, the sample size is still huge as it represents 20 percent of the whole 40 year period. Not even does the higher percentile not support the previous highlighted connection, it further suggests that T2m is leading the SST, raising the question how reliable the claim is that the cold Atlantic SSTs can be seen for a precursor of warm European temperature anomalies. Additionally, this includes the description of lines 226-240 (Fig. 6), the authors should give further interpretation to these correlation plots. It seem a bit surprising that for the cold and warm cases correlation with T2m goes towards 0 towards higher positive lags (stronger for higher percentile in Fig S2) or positive for negative lags, whereas the climatological signal is strongly negative (at high positive lags) or has the opposite sign for negative lags. As for Fig. 6, warm and cold cases already represent 50 percent of the data, this means that the more climatological cases (in SST anomalies) should represent strong negative correlations (?). Wouldn't this somehow contradict the importance of this dynamical connection associated with strong Atlantic SST anomalies? What about cold/warm cases in T2m (instead of SST), if those would be included in the correlation, would they agree with the given interpretation for the importance/impact of SST anomalies?

**line 249:**
I think the SST anomaly was previously not refered to as subpolar. Maybe it makes sense to be consistent in using the same terms, as subpolar probably is misrepresenting the very specific cold/warm blob anomaly in the Atlantic.

**lines 251-253:**
I think I mentioned this earlier already, but isn't for the QRA a more global double jet structure necessary whereas here the author are limiting this to a very regional feature? So if the authors want to limit their analysis to a regional double jet structure, why do they pick specifically the QRA as underlying wave response, or if they want to investigate the impact of the QRA, why limiting the double jet structure locally?

**line 254:**
Maybe refering to the "extended" summer season, as MJJA is not the usually used range for summer season.

**lines 258-259:**
Could the authors include in their correlation diagnostics maybe also some estimate for statistical significance, something like combining correlation with associated p-values? Also how much depends this picture (Fig. 7) on the choice of the DJI, which seems (as previously mentioned) to be chosen subjectively (or is this a usual way to measure the DJ structure?). But anyway, in general it would be interesting to know how sensitive the results are on the choices for the DJI (to define the index as well as the chosen longitude range) or maybe as well the SST average area (what about the second, newly introduced slightly shifted area?).

**caption Fig. 8:**
figure caption only for upper panels, not lower panels. Further, how do the quantiles match? (title 0.25-0.75 quantiles, caption 10% most extreme cases). And what data is actually used, I am not sure I fully understand the description, mentioning the coldest 10% with the 10% highest DJI. Are the anomalies calculated from the dates either being in the 10% coldest cases or 10% highest DJI, or in both highest 10%? I think this need to be made clearer.

**lines 272-274:**
Coming back to one of my previous point about the correlation patterns of Fig. 4 and 5 (as well as Fig. 6). The way I understand those correlation patterns, they show very low frequency variability in all fields, low SST anomalies leading to persisten high pressure or high temperature anomalies, indicating a stationary or quasi-stationary wave response. Even Fig. 6 seems to indicate very persistent T2m/GPH responses of at least 2 weeks. How does this agree with the findings shown in Fig. 8, which indicate that the relevant connection comes from waves with phase speeds 5-10 m/s. In terms of more general definition of waves, this could be explained by propagating and breaking waves,

transferring wave activity to a lower wavenumber wave or a localized blocking high, but how would this agree with the persistent pattern of Fig. 4?

**lines 276:**
Is this true, does a DJI triggers QRA events and are not only a necessary condition for QRA. But apart from this, why is this really surprising, since a very local DJI cannot really be linked to the global QRA mechanism.

**lines 280-281:**
I cannot follow the authors motivation here why they separate the distribution into troughs and ridges occuring over the SST box. What are the reasons for doing this? Following their previous results, cold SST are rather associated with a trough signalin this region. So it would be interesting to know for their separation of how much percentage of the time there is a ridge or trough. From the kind of distribution shown in Fig. 9, this cannot be derived. And wouldn't it also be helpful to show the distribution for the troughs or ridges during this cold SST period instead of the climatology already given in Fig. 8? I assume, the overall distribution looks a bit different, as there are anomalies that are even larger than the underlying climatological distribution (most obvious for w6). Additionally, what do the authors mean by the trough is slowing down the eastward propagating waves. A mechanism of slowing down the wave cannot be derived from Fig. 9, correct? And the trough is part of the wave, so if there is a propagating wave moving across the SST box I would assume that they had more or less the same phase speed if they both are part of the same wave.

**line 287:**
Maybe I misinterpret the plots, but how can the difference plots between troughs and ridges can be used to identify an imprint of high-amplitude waves? The figure doesn't show the underlying distribution of either case, ride or trough. How can it therefore be used to identify the occurrences of high-amplitude waves?

**lines 289-290:**
How does this statement makes sense, that the authors are including also medium-scale waves (4-8), whereas Kornhuber et al. (2017) only investigated wave numbers 6-8. So they also included medium-scale waves. I think the authors should be clearer here on what they mean by that.

**lines 294-295:**
How can Fig. 10 be seen as a measure of wave amplitude? If I understand it correctly, Fig. 10 shows some sort of ridge/trough GPH anomaly for which the amplitude strongly depends on the position of the trough/ridge relative to the SST box. This is not a measure wave amplitude.

**lines 303-305:**
Isn't this a statement about the general deviation of the climatological pattern: increase

in trough associated with lower SST, decrease of the trough higher SST. I cannot really see the link to the general associated wave patterns, above all not for a link to propagating waves.

**2.4 Section 4 (Conclusions)**

**lines 329-331:**
This statement does suggest a strong causal relationship between precursor jet stream and resulting wave pattern. But the authors did not show this causal relationship. The DJ structure could be just an imprint of the underlying wave pattern with a blocking flow feature over Europe, therefore maybe a stronger Jet tilt over the Atlantic with possible increase U values south of it where no blocking flow is present (northward shift of high geopotential height values). I would say that this point was not shown convincingly.

**lines 332-334:**
As mentioned in a previous comment, I think the authors should clarify some points associated with this conclusion and make their point clearer how they get to this conclusion.

**lines 335-337:**
The authors deviate in the method/description for QRA to Kornhuber et al. (2017) and their DJ structure is very regional restricted, it is therefore not so simple to make this link or relate their results to this mechanism and derive conclusions from it.

**lines 338-341:**
As mentioned in my previous notes, I think this needs further clarification and better description.

**References**

A. Duchez, E. Frajka-Williams, S. A. Josey, D. G. Evans, J. P. Grist, R. Marsh, G. D. McCarthy, B. Sinha, D. I. Berry, and J. J.-M. H. Hirschi. Drivers of exceptionally cold north atlantic ocean temperatures and their link to the 2015 european heat wave. *Environmental Research Letters*, 11(7):074004, jul 2016. doi: 10.1088/1748-9326/11/7/074004. URL `https://doi.org/10.1088%2F1748-9326%2F11%2F7%2F074004`.

K. Kornhuber, V. Petoukhov, S. Petri, S. Rahmstorf, and D. Coumou. Evidence for wave resonance as a key mechanism for generating high-amplitude quasi-stationary waves in boreal summer. *Climate Dynamics*, 49(5):1961–1979, 2017. doi: 10.1007/s00382-016-3399-6. URL `https://doi.org/10.1007/s00382-016-3399-6`.

G. Wolf, D. J. Brayshaw, N. P. Klingaman, and A. Czaja. Quasi-stationary waves and their impact on european weather and extreme events. *Quart. J. Roy. Meteor. Soc.*, 144(717):2431–2448, 2018. doi: 10.1002/qj.3310.

---

## Author Comment (AC1) · 17 Nov 2020

**Authors response - North Atlantic SST relationship with Jet Stream anomalies and European Heat waves - A composite study**

Julian Krüger[1], Robin Pilch Kedzierski[1], Karl Bumke[1], and Katja Matthes[1,2]

[1]Marine Meteorology Department, GEOMAR Helmholtz Centre for Ocean Research Kiel, Kiel, Germany.
[2]Faculty of Mathematics and Natural Sciences, Christian-Albrechts-Universität zu Kiel, Kiel, Germany.

**Correspondence:** Julian Krüger (jukrueger@geomar.de)

We truly appreciate and are thankful for the effort that has been put by the reviewers in carefully reading and reviewing our manuscript. By responding to their detailed and constructive comments we believe that the manuscript can benefit substantially. We hope that all their concerns have been duly addressed. Below are our responses to the reviewer, which are implemented in the revised manuscript. The line numbers/figure references in the reviewer's comments refer to the original manuscript.

5    Reviewer's comments are reprinted in a thinner and *italicised* style, our response is typed below it in a **thicker and non-italicised** style.

We will attach the revised manuscript. We do not highlight the changes here, as major re-structuring was done for the revised manuscript.

**1. Review**

**1.1.**

*The current work considers anomalies of SSTs and the upper tropospheric waviness in the North Atlantic sector in an attempt to estimate their impact on the occurrence of European heat waves. Specific aspects of both factors have been suggested in the past to be conducive to European heat waves, but this paper considers, for the first time, both factors simultaneously. A novelty hereby is the fact that the authors focus on the waviness over the North Atlantic rather than the circumglobal waviness as was common in numerous previous studies. The manuscript contains a number of interesting aspects, like the lag-correlation analysis in order to tease out a cause-and-effect relationship or the discrepancies with respect to previous work regarding the wave behavior in connection with European heatwaves. There seems to be potential for progress in these directions. At the same time, I see a number of fundamental weaknesses. First of all, the local double jet index introduced by the authors seems to be incompatible with the idea of quasi-resonance and, therefore, not an adequate metric to quantify the latter. Also, the analysis is mostly based on composites and correlations. A correlation does not imply causation, and therefore the analysis does not live up to the expectation raised by the title ("Impact of .... on ...."), which promises to shed light on causal connections. In addition, many of the shown correlations are pretty weak; in no case do the authors investigate whether the correlations and the patterns on the composite maps are statistically significant or not. Throughout the text the authors relate their results to those of earlier studies, and they sometimes find agreement and sometimes disagreement. In the end this left me lost and I could not find out what I am supposed to learn: are some of the earlier results wrong (because of the detected disagreement), or are the authors' own results an out- come of an inadequate metric (see above) or pure chance (which would be consistent with a potentially missing statistical significance) or both? Below I provide further details under "major issues" in order to illuminate my concerns. Items 3–5 could be addressed by major revisions. Item 2 seems critical, although one might argue that a subsequent proof of statistical significance (if that turned out to be possible) would save the day. However, item 1 is fundamental in the sense that a substantial part of the analysis is based on the local double jet index, which in my eyes is inappropriate to fulfill its purpose. Addressing all these issues will basically result in a very different manuscript. I thus have to recommend rejecting this manuscript. I chose not to mention further minor issues.*

**Many thanks for the helpful comments. We are aware now that the language choices specially at the beginning of our manuscript were perhaps implying more causal and/or mechanistic content than what's actually shown in the manuscript. We've made a major effort to improve the accuracy of the terms used in the revised version (see details in the point-by-point replies below).**

**In the previous manuscript version, too much weight was given to the QRA throughout the text and again specially at the beginning, giving the impression that the analyses deviated from the initial 'goal' spontaneously.**

**We want to highlight that our wave analysis (Figs. 8 and 9) with wave phase speed and wave amplitude distributions, is**

not specific to the QRA: those distributions give excellent insight into what kind of waves dominate our composites: stationary or travelling; near-normal or amplified waves. We make sure to introduce our wave analysis in a more general way now so the reader keeps an open mind throughout the text, specially given that the wave analysis sections comes last.

The main conclusion of our study is that the relation of higher occurrence of ridges (including heatwaves) over Europe, with lower SSTs in the North Atlantic, involves travelling waves which are not necessarily amplified. We feel it is a relevant conclusion for the community, given that a lot of current research concentrates on stationary atmospheric response to SST anomalies.

In the previous manuscript version, the narrative was, in short : 'We were looking for the QRA, instead found something else'. We've made an effort to introduce and contextualize better this 'something else', apart from relating it to known literature about the recurrent Rossby wave pattern (Röthlisberger et al., 2019). It is clear this was not highlighted and discussed properly enough before.

Further we agree that statistical significance measures were missing in our study. We took this into account and implemented different measures to confirm the reliability of our results (e.g. Fig. 4 - standard deviation; Fig. 7 - correlation coefficient and associated p-value).

The correlation analysis did not convincingly show a distinct lead-lag relationship. Instead of showing a correlation based on day-to-day variability, we now show the lead-lag relationship with whole distributions, that more easily captures whether one of the tested parameters is leading the other or not.

Additionally we produced a circumglobal DJI measure, that qualitatively give a similar result compared to the locally restricted DJI measure. For a detailed description we refer to our reply of the comment below.

*1.2.*

*The theory of "quasi-resonance" requires Rossby waves to travel around the en- tire globe. This is a fundamental aspect of quasi-resonance, because the singularity in the case of no damping (see equation (3) in Petoukhov et al. 2013) results from the constructive interference of an infinite number of waves, which is only possible if the domain is periodic in longitude and if the waves see a zonal waveguide at all longitudes. Correspondingly, previous authors have analyzed the zonally averaged zonal wind and computed the corresponding stationary wavenumber $Ks(\phi)$, which depends on latitude $\phi$. According to WKB theory, a relative maximum in $Ks(\phi)$ — which typically occurs in the presence of a double jet structure — is then indicative of a zonal waveguide all the way around the globe. In the current manuscript, the authors diagnose the double jet structure locally over the North Atlantic ocean. This leaves open two issues: first, does this also correspond to a relative maximum in $Ks(\phi)$? And, second (more fundamentally), even if there were a local (in longitude) relative maximum in $Ks(\phi)$, this alone would not indicate a wave guide all around the globe. Therefore, it seems to me that the authors' local double jet index is inadequate for diagnosing a zonal waveguide as required for the validity of quasi-resonance theory.*

It was not the goal of our paper to perform a full diagnostic of WKB theory. As we point out now in the introduction, WKB theory has some limitations (Wirth, 2020), and there are types of Rossby waves other that stationary and

**amplified that could be at play (e.g. the recurrent Rossby wave pattern that involves travelling waves). Our approach**

80 **is more general in order to find out which wave kind is dominant in our composites. We've also increased the depth of our discussions in order to relate our results more clearly to previous literature.**

**We've performed a circumglobal measure of the DJI with qualitatively similar results, please see the bottom panel of Fig. 7 in the revised manuscript version. This makes sense given that, climatologically, the North Atlantic – European sector is the one showing a double-jet structure most markedly (Hall et al., 2015), dominating to some degree the zon-**

85 **ally averaged measure. We've tried to separate more clearly the part of our manuscript related to precoditioning of the wave; and the actual wave analysis (Figs. 8 and 9) that shows the imprint of the dominant wave type.**

**Although we already mentioned that similar results were achieved using SST, DJI, or both combined for compositing wave properties in what's the current Fig. 8, for consistency we only use SSTs throughout the main manuscript, pointing out to the supplement to show the similar result using DJI combined with it.**

90 **We advocate that this major comment is not so fundamental in essence, given that the main conclusions of our paper would not change or go missing even if we eliminated all results involving DJI from the manuscript.**

*1.3.*

*One key concern is that many of the shown effects (correlations, composites) are rather weak and it is not clear to me whether*

95 *they are statistically significant or not. It appears necessary to me to determine the statistical significance of all these results before they can be presented and discussed in a scientific paper. For instance: Are the differences shown in Fig. 3 statistically significant? Are the weak correlations shown in Figs. 4 and 5 statistically significant? The same for the correlations shown in Fig. 6, which are even smaller (a correlation coefficient of the order of r = 0.2 corresponds to a "described variance" of r2 = 0.04 = 4%, which is very low and requires a large and high-quality data set to tease it out in a significant manner). The issue*

100 *of statistical significance and the interpretation of a correlation in terms of causality can be illustrated in Figure 2. Apparently, some heatwaves over Europe are associated with cold SSTs, some are not, and sometimes there is a cold SST anomaly but no heat wave. Overall this is a very mixed bag, and I believe that the correlation between two respective time series is very low and probably not significant. On the other hand, if cold North Atlantic SSTs were really "drivers" ("impact", "influence", etc.) of European heatwaves, wouldn't this correlation have to look more convincing?*

105

**Thank you for this comment. Regarding Fig. 2 of our original manuscript and Fig. 3 of our revised manuscript, the time series of North Atlantic SSTs and European surface temperatures explains that negative SST anomalies of more than -1°C occurring at leat in one of those two box averages are only present during 1994, 2015 and 2018, which are known as European heat wave years. Consequently, we note that at least strongly negative North Atlantic SSTs are**

110 **all related to European heat waves. This is the main result of the figure and we investigated how these parameters are linked together by using a composite of cold SST anomalies as well as studying the case studies. Here we do not pose the other way around, i.e. that each European heat wave must be accompanied by negative North Atlantic SST anomalies. We added a statistical significance measure to make the results from the Figs. 3 and S1 in our original manuscript more**

**reliable, which are now found as Fig. 4 and S2 in our revised manuscript. The standard deviation of the medium SST**
115 **events show that the deviations during cold and warm SST events are significant for both chosen quantiles.**
**The correlation analysis including Figs. 4,5, and 6 is removed from our study, as it's better to show the full distributions,**
**rather than a correlation coefficient that describes linear day-to-day relation in a non-linear system.**

*1.4.*

120 *The authors talk about "impact of ... on ...." (title), about "drivers of ...." (second paragraph in the introduction), about the*
*"influence on ...." (section 3.1 head- ing), "A promotes B...." (conclusion section) etc. All these terms suggest that this paper is*
*about identifying causes for European heat waves. Correspondingly I started to read the paper with high expectations and was*
*disappointed later on, because in my eyes the kind of analysis presented in the manuscript does not allow one to determine the*
*causes for heat wave occurrence. As I said above, correlation does not imply causation. More specifically, I do not believe that*
125 *the "influence of ... on ..." can be "evaluated by using a composite study" (as promised, e.g., on line 96). An analysis based on*
*composites or on correlations is at best suggestive, but certainly not conclusive. To give an example: the plot shown in Fig. 3*
*indicates an association between North Atlantic SST and the upper tropospheric flow over that region in two particular years.*
*The questions that remains unanswered: are these two years just a fortuitous coincidence (see my earlier issue about statistical*
*significance), does the ocean "drive" the atmosphere, or does the atmosphere "drive" the ocean? In summary: given what the*
130 *authors show in the manuscript, I think that the language is inflated and the analysis does not live up to the expectations raised*
*by some of the formulations.*

**Thank you for this advice. We will be careful about suggesting a causal relationship and rephrased several para-**
**graphs without proposing a sequence of responses. Therefore we also modified the title of the study and will not talk**
135 **about an 'Impact of ... on ...' anymore, but rather about a tested relationship by using a composite study.**

*1.5.*

*Some of the reasoning seems rather casual. For instance, the authors extend the analysis of a previous study to the year 2018*
*and show that the European heat wave was associated with a simultaneous cold SST anomaly over the North Atlantic similar*
140 *as in that previous study for a previous year. So far so good. On the other hand, Fig. 2 shows that some other extreme heat*
*waves such as the 2003 or the 2010 heat wave were not associated with strong cold SST anomalies over the North Atlantic. So*
*what am I supposed to learn from that? Does it mean that the association between SST anomalies and European temperatures*
*is a random event? The authors do not provide any guidance to the reader as to how to interpret these results.*

145 **As already written in comment 1.3, we highlight that anomalously cold North Atlantic SST anomalies are associated**
**with European heat waves and we do not claim that each heat wave year is accompanied by or has negative SST anoma-**
**lies in advance. For instance, the heat wave during 2010 does not show negative SST anomalies, but during that summer**
**rather eastern than central Europe was affected by enhanced warmth. We further support our hypothesis by including**

**the year 1994 in our case study, where a strong anti-correlation is found.**

150

*1.6.*

*I have an issue with the lag correlation analysis (Figs. 4, 5, and 6) in combination with preprocessing the involved variables through a 15-day running mean. A running mean is a non-causal filter which projects information backwards in time. I do not say that the results shown in Fig. 6 are invalid, I am just skeptical and it would be more convincing if the same results could*

155 *be obtained without a non- causal filter. Also, the four plots in Figs. 4 and 5, respectively, are very similar to each other, and it appears to me that this is in essence a result of the 15- day running mean filter preprocessing: the latter smears out the data in the time domain such that a time lag of 4 or 8 (or even 12) days becomes almost invisible. In the end the issue goes back to the question of statistical significance (see one of my previous items). For instance, the effective number of degrees of freedom in Fig. 6 is likely to be very low as a result of the 15-day running mean. This makes me believe that a difference in the correlation*

160 *coefficient of as little as 0.2 between a 15 day lead and a 15 day lag could be statistically insignificant.*

**We removed the entire correlation analysis, as this style of lead-lag analysis only investigates the day-to-day relation. On top of that, we found a mistake making the correlation plots shown in the previous manuscript version less reliable. We removed this part, but instead showing a lead-lag relationship including the full distribution shifts (Fig. 6 of the**

165 **revised manuscript).**
* * *
2. Review

*2.1. General comments.*

170 *The submitted manuscript covers important dynamical aspects on the evolution of heat waves and tries to link those to understand the underlying mechanisms. The authors focus on previously identified potentially important drivers of heat waves, namely the wave amplification according to the quasi-resonance mechanism and Atlantic precursor SST anomalies. Those feature have been previously shown to be linked to some heat waves. The authors further investigate the role of jet stream anomalies as link between those features and try to answer the question about their importance on heat waves. The study*

175 *nicely combines the investigation of two case studies (heat waves in 2015 and 2018) and a more general climatological perspective of a 40 years period using ERA5.*
*The study is nicely motivated and it picks up two (three) mechanisms which are further investigated in detail and the authors tried to investigate a possible link between them. This setup of the paper appears quite exciting, however, the manuscript still needs some substantial improvement to more convincingly support the claims they present with their results. One major issue I*

180 *have with the study is that their quick conclusions about causal relationships is not made very convincing. I don't think that the presented results are enough to support their causal relationships (specific comments about this further below). This seems to be even more problematic that the results are not supported by any means of statistical significance test. As mentioned further below for the specific comments I did struggle to understand if the shown patterns are really meaningful enough to come to the*

*conclusions by the authors. I think applying some measure of statistical significance could be helpful for the authors to make*
*their point more convincingly and convince the reader that their results are indeed meaningful. A related issue I have with this*
*is the missing understanding of sensitivity of the results to some of their choices. I made this point also in the specific com-*
*ments, like for example for the choice of double jet index. I would suggest that the authors should either refer to another study*
*where this index has been used, or at least motivate their choices or tell the reader about the sensitivity of the results on those*
*subjective choices. This issue appears several time during the manuscript and is also enhanced by the authors themselves, by*
*showing that changing the averages boxes for SST or temperature can lead to different results. This raises the question, how*
*much the following results can be trusted or if a meaningful interpretation is difficult due to this sensitivity. I would suggest*
*that the authors give the reader further insight into the impact of these changes.*

*Further, I did struggle a bit to understand what the actual real goal of this paper is, related to wave amplification mechanisms.*
*The quasi-resonance mechanism is a possible mechanism relevant for heat waves, but it seems that the authors are interested in*
*the dynamics between different contributors in the Atlantic to support the evolution of heat waves, SST anomalies, jet structure*
*and evolving wave patterns. This impression is further strengthened by the authors choices to deviate from the procedure of the*
*quasi-resonance mechanism on a more locally restricted scale (in terms of extension of double jet structure). But why then not*
*using a more general wave measure? Previous literature has shown that not all heat waves are associated with global scale*
*waves, but more locally restricted ones. I assume the reason is that the authors were only interested in this sub-class of waves*
*associated with heat waves, which is fine, but I think that the authors then should make this point more clearly and highlight*
*that they specifically are only focusing on such waves. This includes also to clarify why the authors nevertheless deviate in their*
*search for dynamical links to locally restricted jet anomalies. So the authors should clarify their choices, that they investigate*
*potentially important aspects that can lead to heat waves, but also why they link those without clarifying why the authors think*
*it is reasonable to link and consider explicitly those features together?*

*In general, I think the manuscript represents a nice study which tries to understand specific aspect and their dynamical links*
*that can lead to European heat waves. However, at the moment the conclusions of the paper are not presented convincingly.*
*This could be improved by better explain to the reader the motivation behind some of their diagnostics, including some support*
*from including some kind of statistical significance test, comments about the sensitivity of the results and being a bit clearer on*
*the aim of this paper (related to the applied diagnostics).*

**Many thanks for the helpful advice regarding our manuscript in general and the detailed review subsequently. We
applied a major revision on our manuscript. Due to the fact that our study doesn't provide a causality between the
North Atlantic SST's and the European surface temperature, we modified the title of the study from 'Impact of North
Atlantic SST and Jet Stream anomalies on European Heat Waves' to 'A composite study of the North Atlantic SST
relationship with Jet Stream anomalies and European Heat waves' among additional rephrasing regarding the causal-
ity originally proposed between North Atlantic SST's and European heat waves within the manuscript, but without
changing the main message of the study.**
**We agree that statistical significance tests were missing in our study, so we applied requested statistical significance**

measures to Fig. 3 and Fig. 7.

220   In order to support the results taken from our composite study and the case studies of 2015 and 2018 (Fig. 3), we also include the case study of summer 1994, where another heat wave occurred over Europe, that was likely to be induced by recurrent Rossby-wave-pattern (Röthlisberger et al., 2019).

Another major point is the specific definition of the locally restricted Double Jet Index. For comparison, we now applied a similar analysis to the zonal mean zonal wind structure and we finally obtain a similar result, showing that

225   higher circumglobal DJI values as well as to the North Atlantic locally restricted higher DJI values are more likely to co-occur with negative North Atlantic SST anomalies. However, we avoid to make a direct relationship between our locally-defined Double jet measure and the QRA-mechanism. Nevertheless the locally restricted DJI measure qualitatively agrees with the zonal mean DJI measure, which makes sense since the double-jet structure tends to be in the North Atlantic-European sector as shown in the jet stream climatologies by Hall et al. (2014)

230   The addressed sensitivity for different measures like the North Atlantic SST box average or the Double jet index is investigated and we conclude that an anti-correlation between the North Atlantic SST and the European surface temperature still holds during European heat wave summers for the two different North Atlantic box averages. The choice of the double jet index is based on the climatological behaviour, however, as above mentioned we additionally performed an circumglobal analysis for the double jet strength. We obtained a similar relationship between the double jet strength

235   and the North Atlantic SST anomalies for both measures.

Furthermore we omit Figs. 4, 5 and 6 as we better want to show the full distributions, rather than a coefficient that describes linear day-to-day relation in a non-linear system. On top of that we found an error in our code, that makes these correlation analysis even less reliable. Instead, we added a lead-lag relationship based on composites to the study.

240   *2.2.*

*Specific comments*

*Introduction*

*lines 60-62*

*The aim of this paper is not fully clear to me and I am slightly confused about the focus between SST anomalies, heat waves*

245   *and QRA. These two sentences are a good summary for this slight confusion, because it seems like the authors have some underlying assumptions that are not explicitly made clear to the reader. According to the second sentence (lines 61-62), the research objective is to investigate the connection between European heat waves and North Atlantic SST anomalies with the double jet structure as possible dynamical link between those two (with the associated wave properties that can be linked to the jet structure). Therefore the authors explicitly want to investigate the link between Duchez et al. (2016) and Kornhuber et*

250   *al. (2017). I find it difficult to understand this reasoning. If I understand the mentioned studies correctly, Duchez et al. (2016) try to link the SST anomaly to heat waves, resulting from some form of atmospheric wave signal that can be linked to the SST anomaly. The QRA proposes a specific subset of waves which are assumed to be important for the occurrence of heat waves. However, the QRA is a subset of waves and there are also more general studies linking atmospheric waves to European heat*

*waves. As the authors are interested in linking SST, jet structure, waves and European heat waves it is not clear why the study*
255 *attempts to use the QRA as the underlying wave mechanism of relevance. E.g. in Wolf et al. (2018) the authors showed on the one hand a possible relevant contribution from the QRA to extreme positive temperature anomalies over Europe during NH summer, but also mentioned on the other hand that a specific contribution of the waves connected with extreme events were a rather local feature which does not so well fall into the QRA perspective. As the study of Duchez et al. (2016) is not depending on the QRA nor did they make this connection, it is not clear why the authors focus explicitly on this link. So the main objective*
260 *is to understand the QRA and trying to identify possible underlying processes for this QRA, for which the authors focus on the SST anomalies highlighted by Duchez et al. (2016)?*

**Thank you for these useful comments and hints as well as for additional references of previous studies, which are closely related to our issue. We tried to make the objective of our study more comprehensible and therefore we restruc-**
265 **tured the paragraph. Additionally we added another mechanism and references in the introduction, i.e. for the studies by Röthlisberger et al. (2019) and Fragkoulidis et al. (2018).**
**We note that our study does not perform the complete QRA diagnostics, as the mechanism explains only a subset of waves. Nevertheless we investigated whether a connection between composited SST anomalies and the double jet as a precondition for the QRA mechanism can be found. We provide the study with correlation coefficients and p-values**
270 **of the relationship between North Atlantic SST's and DJI values. We found a correlation of -0.2587 with a p.value of 0.0008 for the locally-restricted DJI and a correlation of -0.1590 with a p-values of 0.0420 for the circumglobal measure. Nonetheless we attenuate the weight of the QRA mechanism in this study as it does not play a major role for the main message of the paper.**

275 *2.3.*
*lines 66-69: The objective of the study is to understand the dynamical link between SST, jet structure and temperature anomalies. Wouldn't it therefore make sense to subtract the climatological trend from the underlying variables to not run into danger to get a bias into the results for more extreme warm events occurring later in the investigated periodic?*

280 **We already show SST time series detrended, and we make it more clear in the methods and results sections, that the detrended data are used.**

[Figure]

**Figure S1.** SST trend (ocean) and 2m.-Temp. trend (land) per Decade based on ERA5 data (1979-2019)

*2.4.*

*lines 74-76: As previously mentioned, I find it difficult to understand on the one hand the focus on the QRA, but on the other*
285 *hand deviating from it to investigate rather more local processes. This somehow does a bit contradict with explicitly focusing*
*on the global QRA feature. The authors should probably be clearer on what their study is aiming for - what is it exactly they*
*try to understand.*

**As above mentioned, we rephrase the paragraph and will not relate our locally restricted DJI measure directly to the**
290 **QRA mechanism anymore. However, we produced another figure (Fig. S2) that includes the zonal mean zonal wind**
**distribution for case studies of MJJA 1994, 2015 and 2018. The jet structure during at least July 1994 and 2018 are in**
**a similar shape to what Kornhuber et al. (2017) found for the cases in 2003 and 2010. Thus, a circumglobal measure**
**based on the latitude ranges, where both maxima and the minimum in between occur in these case studies is created.**
**The newly defined circumglobal MJJA DJI values show a similar distribution in a scatter plot against the North Atlantic**
295 **SST anomalies: Higher monthly DJI values are more likely during negative SST anomalies, particularly most of the**
**July values of the European heat wave years (1994, 2003, 2015 and 2018) are found in the upper left corner representing**
**strong double jets co-existing with negative North Atlantic SST anomalies. We note with this figure, that the distribution**

of the double jet values with respect to the North Atlantic SST is similar to the distribution found in the original figure, where a locally restricted DJI measure is used. We conclude that the North Atlantic dominates the zonal-mean double

300   jet presence, as the spiral-like structure of the jet stream is centered there in the climatological state (Hall et al., 2014). We provide this figure here to show that our defined DJI is reasonable. We refrain from replacing the figure in our manuscript, as the wave analysis is based on the DJI values produced with the locally restricted double jet measure..

[Figure]

**Figure S2.** Zonal mean zonal wind distribution for July 1994, 2015 and 2018. Dashed lines are denoting the latitude bands, where the poleward (60 and 80°N) and the equatorward (30 and 60° $N$) maximum and the minimum (50 and 70 °N) in between will be found.

[Figure]

**Figure S3.** North Atlantic SST anomaly against a circumglobal measure of the double jet strength, defined as the sum of maximum zonal mean zonal wind speed found between 60 and 80°N and 30 and 60°N and subtracting minimum zonal mean zonal wind strength found between 50 and 70 °N.

2.5.

*lines 82-83: In the previous paragraph the authors explained how they identify the three climatological branches of the jet structure for May/June and July/August from the climatological state. So the branches define these three step functions (black lines with dots) in Fig. 1, correct? In lines 82-83 the authors explain how they calculate the branches for each month, but I thought the branches are derived from the climatological fields. Do the authors intended to explain how they calculate the associated U-values for each month based on the climatological branches? Maybe the authors could make this description a bit clearer.*

This part needs some more description on how the analysis is done step-by-step. Thank you for this advice. The Fig. 2 in the revised manuscript illustrates the climatological state for two bimonthly means, respectively. Based on the bimonthly state, we defined the three branches. These branches are then applied on each month within the respective bimonthly time range.

315 The U-values on each of those three branches (upper maximum, lower maximum and minimum in between) are then averaged for each month. Consequently, we obtain three values(strength of poleward maximum, equatorward maximum and the minimum in between). In order to relate this to the climatology, we divided the result of (U poleward+U equatorward - U minimum) for each month by the climatological value of this sum. Hopefully these lines and particularly the modified description in the manuscript is sufficient to clarify the different steps of the analysis.

320

*2.6.*

*line 94: Is there a reasoning behind the specific choice of DJI or do the specific not really matter? It is some kind of curvature (Unorth - 2\*Umid + Usouth), but with more weight to the absoulte maxima of the jet maxima. Would the use of the curvature instead have an impact on the results?*

325

The intention was to define a measure for the double jet strength, which is relative to the climatological data. With the suggested calculation of a curvature (Unorth-2\*Umid+Usouth), we obtain a similar picture, but with an amplification of the values relative to the climatology, i.e. smaller values get smaller, stronger values get stronger. Thus, we think that using this measure, would not be more beneficial or impact any of our conclusions, so we retain the original figure in 330 our manuscript.

[Figure]

**Figure S4.** original figure (Unorth - Umid +Usouth)

[Figure]

**Figure S5.** use of curvature (Unorth-2*Umid+Usouth)

*2.7.*

*line 97: How do the authors remove the seasonal variability? Do they use a daily based climatology? I think the authors should specifically mention this.*

**Thank you for announcing this missing information in the manuscript. The seasonal variability is removed by using a daily-based climatology. We added this information in our manuscript.**

*2.8.*

*lines 104-105: What does this exactly mean (daily-based detection of the jet stream)? Is the maximum calculated from the composite fields and daily based refers to applied timelags for this composite, or is the location of the maximum represented by the average of the locations for each maximum at every day used for the composite?*

**We try to explain this here as well as in the manuscript more comprehensively: First we obtain the SST composites. At the dates of the respective SST composites, we search for the jet maximum with daily values. These are averaged, so each composite gain one jet stream maximum structure, shown in Fig. 3a.**
**This analysis doesn't include time lags, as this could be understood from our present description.**

*2.9.*

*line 110 (but also valid for the other figures): I would suggest to order the figures in the way they are referred to in the text.*

**Re-structuring of the manuscript was part of the major corrections performed in the study, so we hope the figure order is more straightforward now.**

*2.10.*

*line 110: Is there a reason to apply a 15 day running mean to one variable, but not the other? Why is this running mean necessary. Using the 0.25/0.75 quantiles (or even 0.1/0.9) should include enough cases to get a smooth line for the time-lagged correlations.*

**We removed the Figs. 4,5 and 6 and therefore also the methods section containing the correlation analysis and also the 15-day running mean. We explain the detailed reasons for the removal below, where additional annotations are made to the correlation analysis section.**

*2.11.*

*line 125: Maybe the authors could spend a bit more time on explaining this index. What does this measure of a trough or ridge, relative to the zonal mean tells the reader. An overall southward or northwards shifted jet in the Pacific would by this measure have an impact on zanom in the Atlantic. This could result in an increase or decrease of the relative strength of the Atlantic ridge or trough, even for identical conditions in the Atlantic. How can this interpreted or what is it aimed for to explain? Also in the following the definition of a trough/ridge seems a bit surprising. Isn't this region in a climatological sense represented by higher z-values, which would mean that by this definition one could identify ridges (zanom¿0.5) for cases when a gridpoint based climatological anomaly would indicate a negative anomaly (trough).*

**Many thanks for these hints. We would like to split these comment into two parts:**

**1.) The reviewer questions the chosen way of calculating the trough/ridge presence relative to the zonal mean. It is adduced that a potential northward/southward shift of the jet stream in the Pacific for instance would lead to a bias in the measure. We would argue that a potential shift of the jet stream in one direction is very likely accompanied by a shift into the opposite direction (conservation of angular momentum) in another longitudinal region, so that a zonal mean would average out these effects. Since we are using the same latitude band for the North Atlantic mean as well as for the zonal mean, this definition could be still be hold in our opinion. Further the use of the Eulerian framework and zonal-mean and eddy terms is the most used wave analysis method. Variations in the zonal-mean are orders of magnitude smaller than the eddy terms.**

**2.) In the second part of this comment the reviewer addresses that the region where the North Atlantic mean is calculated is in a climatological state represented by higher GPH values. The studies by Hall et al. (2017) and Woollings and Blackburn (2012) support that the investigated region (45 - 60°N, 14 - 40°W) is part of the North Atlantic southwest-northeast tilt, so the region is climatologically located neither in significantly positive nor in significantly negative GPH anomaly values.**

**The climatology is actually shown in the plot, so the reader can see the shift in trough/ridge occurrence under different SST conditions, relative to the climatology.**

**We will simplify the measure, so that it is explains the difference between the North Atlantic mean and the zonal mean directly in the unit of geopotential height (gpm) instead of %.**

390

*2.12.*

*2.3 Section 3 (Impact of North Atlantic SST anomalies) lines 155-159: What is the conclusion here? Must the hypothesis be rejected that cold Atlantic are leading/associated with European heat waves? Or is it necessary to analyze the spatial pattern shift in T2m and SST anomalies? The authors should be clear on their interpretation of their results. This missing interpreta-*

395 *tion also raises further questions in the following paragraph. The authors indicate that there is no clear connection between Atlantic SST and European T2m, but now they investigate if the SST is responsible for the phase-locking of the jet leading to summer extremes in Europe, which they said are not linked. I therefore suggest that the authors include some further description and interpretation of their results to avoid this confusion.*

400 **Many thanks for this comment. We need to mention here and in the manuscript that the major heat waves over central Europe (1994, 2003, 2015 and 2018) are all showing the same feature during summer season (JJA): mostly strong negative North Atlantic SST anomalies are co-occurring with positive 2m- air temperatures. This pattern during the heat wave years over central Europe must be highlighted. Therefore we are interested to make an analysis about how these parameters could be linked together. We will rephrase the paragraph to prevent possible confusion, that seems to**

405 **evolve through the comparison between the two different box averages.**

*2.13.*

*lines 191-192: How do the authors come to this conclusion? Jet stream position/wind and SST anomalies are maybe not surprisingly associated, therefore a change in the jet stream position will be also associated with SST anomalies, or those SST*

410 *anomalies could also be driven by a shift in the jet stream position. The authors have not investigated the causal relationship and therefore should rephrase their statement that it does not lead to a misleading interpretation of a causal relationship. The author should be clear on this and rather explicitly refer to their following lead-lag-correlations to gain further insight into this connection.*

415 **Thank you for this important remark. We were now careful throughout the manuscript to avoid implying any causal relationship.**

**Given our new results with lead-lag distribution anomalies, we state now that a conclusion cannot be obtained of which of SST and Jet stream is forcing the other. Another possibility is a two-way interaction or feedback, which would need further study.**

420 **Also, we've increased the discussion depth providing literature about SST and jet stream anomaly forcing mechanisms.**

**We want to highlight that the observed Jet Stream anomalies during summers with cold North Atlantic SST's make sense in light of current research.**

**We will replace particularly the word 'responsible', as we don't proof that the North Atlantic SST is responsible for a certain jet stream pattern.**

*2.14.*

*lines 202-207: These findings (Fig. 4) are a bit surprising, maybe the authors can give some further insight and interpretation of these findings. With synoptic systems usually dominating the midlatitudes (and representing the GPH fields), I would not necessarily expect such a persistent correlation pattern. As the correlation maps look nearly constant, I would also expect the same correlations for 20 to 30 days. This however would make me doubt this is a good measure to investigate the prediction of some dynamical features as the build up of a wave pattern. And does the correlation goes towards zero for negative lags? I have the same issue also with Fig. 5. The patterns suggest some longterm connections, like large scale NAO patterns which can be associated with large scale flow and temperature anomalies. But to investigate such features, a 4 day lag seems a bit strange, which would rather suggest the authors try to investigate the quick atmospheric reply to the cold SST anomalies in amplifying the wave pattern downstream. I think the authors should better describe what it is exactly they are trying to investigate and with it explain their choice of time lags. Do those kind of plots show that the SST anomalies are extremely persistent and define the overall large scale flow pattern for Europe for the whole summer. If so, shouldn't the authors also show negative lags to support their statement of a causal relationship? I could also imagine a wave pattern building up the SST anomalies, with the wave pattern amplifying itself further, which could also lead to the shown correlations, but which would also have the same correlation pattern for negative lags.*

**We have to mention that we removed the correlation analysis from our manuscript, as we uncovered deficiencies regarding the meaning and the significance of the correlation pattern. The reviewer already mention that the correlation pattern should, but do not cover the dynamical aspects in order to explicitly determine a prominent lagged relationship between each of those tested parameters. Our correlation coefficient describe a linear relation of day-to-day variability between the lower frequently varying North Atlantic SST and the more higher frequently varying 2m-air Temp./ 300 hPa GPH in a non-linear system, so the correlation coefficients are not the best suited for a lead-lag analysis.**

**Further the higher percentiles don't show convincingly that the North Atlantic SST could lead the 2m-air temperature (Fig. S2 in the supplement).**

**Therefore we removed the Figs. 4,5 and 6 and S2 from our manuscript. The key message of our manuscript is still valid without the correlation analysis, as we extended our wave analysis with an investigation of a lagged relationship in a different style. The Fig. 6 in the revised manuscript explains the lagged relationship between the trough/ridge occurrence specifically in the North Atlantic an Europe during cold North Atlantic SST events (0.1 quantile), so it rather shows composited distribution anomalies over different lags.**

*2.15.*

*lines 241-243: I think more discussion is necessary and discrepancies cannot be disregarded due to a small sample size. First, if the mechanism is relevant, it should also be visible in the more extreme cases - I thought this is the whole point, presenting a mechanism for extreme summer events. Second, the sample size is still huge as it represents 20 percent of the whole 40 year*

460   *period. Not even does the higher percentile not support the previous highlighted connection, it further suggests that T2m is leading the SST, raising the question how reliable the claim is that the cold Atlantic SSTs can be seen for a precursor of warm European temperature anomalies. Additionally, this includes the description of lines 226-240 (Fig. 6), the authors should give further interpretation to these correlation plots. It seem a bit surprising that for the cold and warm cases correlation with T2m goes towards 0 towards higher positive lags (stronger for higher percentile in Fig S2) or positive for negative lags, whereas the*

465   *climatological signal is strongly negative (at high positive lags) or has the opposite sign for negative lags. As for Fig. 6, warm and cold cases already represent 50 percent of the data, this means that the more climatological cases (in SST anomalies) should represent strong negative correlations (?). Wouldn't this somehow contradict the importance of this dynamical connection associated with strong Atlantic SST anomalies? What about cold/warm cases in T2m (instead of SST), if those would be included in the correlation, would they agree with the given interpretation for the importance/impact of SST anomalies?*

470

**Here, we refer to the response to the comment 2.14., that explains why we removed the correlation analysis including the Figs. 4, 5, 6 and S2. from the original manuscript.**

*2.16.*

475   *line 249: I think the SST anomaly was previously not referred to as subpolar. Maybe it makes sense to be consistent in using the same terms, as subpolar probably is misrepresenting the very specific cold/warm blob anomaly in the Atlantic.*

**We corrected it in order to be consistent using the same term ('North Atlantic SST anomaly') throughout the whole manuscript.**

480

*2.17.*

*lines 251-253: I think I mentioned this earlier already, but isn't for the QRA a more global double jet structure necessary whereas here the author are limiting this to a very regional feature? So if the authors want to limit their analysis to a regional double jet structure, why do they pick specifically the QRA as underlying wave response, or if they want to investigate the*

485   *impact of the QRA, why limiting the double jet structure locally?*

**In the revised manuscript we provide a zonal-mean measure of the double jet structure. A more extended reply is found in comment 2.4. above.**

490 *2.18.*

*line 254: Maybe referring to the "extended" summer season, as MJJA is not the usually used range for summer season.*

**Thanks for this hint. We will refer to an extended summer season, as it includes one additional month.**

495 *2.19.*

*lines 258-259: Could the authors include in their correlation diagnostics maybe also some estimate for statistical significance, something like combining correlation with associated p-values? Also how much depends this picture (Fig. 7) on the choice of the DJI, which seems (as previously mentioned) to be chosen subjectively (or is this a usual way to measure the DJ structure?). But anyway, in general it would be interesting to know how sensitive the results are on the choices for the DJI (to define the*

500 *index as well as the chosen longitude range) or maybe as well the SST average area (what about the second, newly introduced slightly shifted area?).*

**We will add a statistical measure to validate the significance of the relationship between the North Atlantic SST and the double jet index.**

505 **Regarding the sensitivity of the DJI values we again refer(Fig. S3, lower panel) and we obtain a similar picture for the overall relationship between the North Atlantic SST and the European 2m- temperature.**
**Regarding the sensitivity of the North Atlantic SST box, we provide here the same scatter plot but with SST box average values of the 'slightly shifted area' (Fig. S6) as suggested by the reviewer, which was shown in Fig. 2a (dashed) of the original manuscript and Fig. 3a of the revised manuscript.**

[Figure]

**Figure S6.** North Atlantic SST (shifted box: 47.5 - 62.5 °N, 20 - 45°W) anomalies versus double jet index (locally restricted based on the branches in the North Atlantic area)

 *2.20.*

*caption Fig. 8: figure caption only for upper panels, not lower panels. Further, how do the quantiles match? (title 0.25-0.75 quantiles, caption 10% most extreme cases). And what data is actually used, I am not sure I fully understand the description, mentioning the coldest 10% with the 10% highest DJI. Are the anomalies calculated from the dates either being in the 10% coldest cases or 10% highest DJI, or in both highest 10%? I think this need to be made clearer.*

515

**We apologize for the mix-up: of course in the caption it should say 25% instead of 10%, as the 0.25-0.75 quantiles are used.**

**For consistency and clarity among section 3.4 in our original manuscript, we moved this figure to the supplement and decided to use only the SSTs for the compositing: as already mentioned in the previous manuscript version, using SSTs,**

520 **DJI, or both simultaneously leads to the same conclusions.**

**We also rephrased the methodology part of the text to be more clear about what data is used.**

*2.21.*

*lines 272-274: Coming back to one of my previous point about the correlation patterns of Fig. 4 and 5 (as well as Fig. 6).*

525 *The way I understand those correlation patterns, they show very low frequency variability in all fields, low SST anomalies leading to persistent high pressure or high temperature anomalies, indicating a stationary or quasi-stationary wave response. Even Fig. 6 seems to indicate very persistent T2m/GPH responses of at least 2 weeks. How does this agree with the findings shown in Fig. 8, which indicate that the relevant connection comes from waves with phase speeds 5-10 m/s. In terms of more general definition of waves, this could be explained by propagating and breaking waves, transferring wave activity to a lower*

530 *wavenumber wave or a localized blocking high, but how would this agree with the persistent pattern of Fig. 4?*

**By staying longer on one phase position (propagating slower within a sector, but not being stationary), the longer-term average is affected, which gives rise to the slight correlation patterns and the composited Jet stream position anomalies. The correlation coefficient is not a good measure at this point, but we can show that the low correlation values already**

535 **hint that the relation of SSTs and 300hPa GPH and 2m-air-Temp. is not a stationary one, which is confirmed with the wave analysis.**

*2.22.*

*lines 276: Is this true, does a DJI triggers QRA events and are not only a necessary condition for QRA. But apart from this,*

540 *why is this really surprising, since a very local DJI cannot really be linked to the global QRA mechanism.*

**We removed this sentence. A strong double jet is a favourable precondition for, but do not trigger the QRA mechanism. With regard to the chosen locally restricted DJI measure, we already explained above the reasoning and performed**

additional analysis with a circumglobal measure, that provides a similar picture as for the local DJI measure.

545

*2.23.*

*lines 280-281: I cannot follow the authors motivation here why they separate the distribution into troughs and ridges occurring over the SST box. What are the reasons for doing this? Following their previous results, cold SST are rather associated with a trough signal in this region. So it would be interesting to know for their separation of how much percentage of the time there*
550 *is a ridge or trough. From the kind of distribution shown in Fig. 9, this cannot be derived. And wouldn't it also be helpful to show the distribution for the troughs or ridges during this cold SST period instead of the climatology already given in Fig. 8? I assume, the overall distribution looks a bit different, as there are anomalies that are even larger than the underlying climatological distribution (most obvious for w6). Additionally, what do the authors mean by the trough is slowing down the eastward propagating waves. A mechanism of slowing down the wave cannot be derived from Fig. 9, correct? And the trough*
555 *is part of the wave, so if there is a propagating wave moving across the SST box I would assume that they had more or less the same phase speed if they both are part of the same wave.*

**As a first option, when discussing Fig. 8 we already explain what imprint would be visible in the wave PDFs if stationary and amplified waves were dominant: positive anomalies around 0 in the x-axis and higher values in the y-axis. Since this**
560 **is not the case in our Fig. 8, one needs travelling waves to explain the time-averaged climatological composite in Fig. 4a) in the revised manuscript, the seasonal averages for specific heatwave years in Fig. 4(b-d) and the increased occurrence of a trough-ridge pair with cold SSTs (Fig. 5).**

**Within the travelling wave kind, two additional options arise: with similar phase speeds, the waves repeatedly amplify more over a preferred position; or with similar amplitudes, the waves spend more time over a preferred position. What**
565 **figure 9 does is to composite the data from Fig. 8 relative to phase position (trough or ridge present over the North Atlantic Box) and show their difference.**

**In Fig. 9 we don't see positive PDF differences between trough-ridge composite in the region of higher amplitudes: the waves don't have increased amplitude when a trough is present over the North Atlantic Box with cold SSTs. Instead, the PDF positive difference is found at lower phase speeds, meaning that during the time-steps in this phase position**
570 **(trough over North Atlantic Box, ridge downstream), the waves are relatively slower compared to the other phase position (ridge over North Atlantic Box, trough downstream). As requested, we show the actual trough and ridge PDFs relative to climatology below: Both show a preference for eastward-travelling waves (as in the overall distributions in Fig. 8), but in the trough composite the waves are relatively slower.**

**Fig. 9 in the main manuscript is the difference between the top and bottom rows.**
575 **Also as requested, the actual occurrence numbers for each composite:**

**Cold SST composite from Fig. 8 $\longrightarrow$ 378 days, of which**

**Trough present over N. Atl. Box $\longrightarrow$ 224 days**

**Ridge present over N. Atl. Box $\longrightarrow$ 35 days**

**As the reviewer points out, one cannot guess the dynamical mechanism by which the travelling wave slows down over a**
580 **certain phase position from our Figs. 8 and 9; however from these distributions it is evident one composite is relatively slower, which is what we are referring to in the paragraph.**

[Figure]

**Figure S7.** Probability density function of phase speed versus wave amplitude (derivation described in section **??**); anomalies are shown for a trough present during cold SSTs in the North Atlantic box (15 - 40°W and 45 - 60°N) (upper row) and for a ridge during cold SSTs within the same North Atlantic box (bottom row); wave properties are shown for 37.5 - 57.5°N for comparison with Kornhuber et al. (2017); black contour lines denote the distribution when cold North Atlantic SST's are present (378 days)

*2.24*
*line 287: Maybe I misinterpret the plots, but how can the difference plots between troughs and ridges can be used to identify*
585 *an imprint of high-amplitude waves? The figure doesn't show the underlying distribution of either case, ridge or trough. How can it therefore be used to identify the occurrences of high-amplitude waves?*

**We refer to the previous reply since it applies to this comment too. We show both trough and ridge distribution above, discussing the meaning of their difference in Fig. 9.**

590

*2.25*
*lines 289-290: How does this statement makes sense, that the authors are including also medium-scale waves (4-8), whereas Kornhuber et al. (2017) only investigated wave numbers 6-8. So they also included medium-scale waves. I think the authors should be clearer here on what they mean by that.*

595

**The reviewer is right that this sentence is quite misleading. We rephrased it and only write in terms of specific wave numbers.**

*2.26*

600  *lines 294-295: How can Fig. 10 be seen as a measure of wave amplitude? If I understand it correctly, Fig. 10 shows some sort of ridge/trough GPH anomaly for which the amplitude strongly depends on the position of the trough/ridge relative to the SST box. This is not a measure for wave amplitude.*

**Thanks for this note. We will remove the phrase of 'wave amplitude' and instead will talk about the 'trough-ridge oc-**
605  **currence.**

*2.27.*

*lines 303-305: Isn't this a statement about the general deviation of the climatological pattern: increase in trough associated with lower SST, decrease of the trough with higher SST. I cannot really see the link to the general associated wave patterns,*
610  *above all not for a link to propagating waves.*

**Something has to lead to the time-averaged climatological composite in Fig. 4a), or the seasonal averages for specific heatwave years in Fig. 4(b-d). Higher occurrence of a trough-ridge pair over a specific location (Fig. 5) necessarily comes from one of three options as described in the the reply to comment 2.23: it's wave anomalies of one kind or the**
615  **other getting integrated over time. Figures 8 and 9 show that, for cold North Atlantic SSTs, it's travelling waves having different speeds at specific phase positions. In restructuring the paper, we introduce trough-ridge occurrence distributions before the wave analysis, as well as explaining Figs. 8 and 9 more in detail now, we hope the current manuscript version is more straightforward to follow.**

620  *2.28.*

*lines 329-331: This statement does suggest a strong causal relationship between precursor jet stream and resulting wave pattern. But the authors did not show this causal relationship. The DJ structure could be just an imprint of the underlying wave pattern with a blocking flow feature over Europe, therefore maybe a stronger Jet tilt over the Atlantic with possible increase U values south of it where no blocking flow is present (northward shift of high geopotential height values). I would say that this*
625  *point was not shown convincingly.*

**We rephrase this part of the conclusion and will 'only' relate or suggest a probable feedback between the mentioned components like North Atlantic SST, double jet structure and wave response. In our study we don't provide evidence, so we will not propose a causality between the components.**

630

*2.29.*

*lines 332-334: As mentioned in a previous comment, I think the authors should clarify some points associated with this con-clusion and make their point clearer how they get to this conclusion.*

635 **As above mentioned, we removed the correlation analysis, so we also delete any conclusion related to these correlation analysis.**

*2.30*

*lines 335-337: The authors deviate in the method/description for QRA to Kornhuber et al. (2017) and their DJ structure is very*
640 *regional restricted, it is therefore not so simple to make this link or relate their results to this mechanism and derive conclusions from it.*

**We reformulate this paragraph, since we don't show that the SST anomalies promote a certain jet stream or wave re-sponse. We only relate the components to each other without testing causality or the sequence of responses.**

645

*2.31.*

*lines 338-341: As mentioned in my previous notes, I think this needs further clarification and better description.*

**Here we agree that we need more clarification, i.e. distinguishing between the overall state when cold North Atlantic**
650 **SST events will happen and the state when additionally a trough is present over the North Atlantic:**
**Cold North Atlantic SST's overall accelerate transient waves (Fig. 8, top row). When a trough is present over the North Atlantic cold SST anomaly, the wave speeds tend to be slower and vice-versa.**
* * *

**North Atlantic SST relationship with Jet Stream anomalies and European Heat Waves - A composite study**

Julian Krüger[1], Robin Pilch Kedzierski[1], Karl Bumke[1], and Katja Matthes[1,2]

[1]Marine Meteorology Department, GEOMAR Helmholtz Centre for Ocean Research Kiel, Kiel, Germany.
[2]Faculty of Mathematics and Natural Sciences, Christian-Albrechts-Universität zu Kiel, Kiel, Germany.

**Correspondence:** Julian Krüger (jukrueger@geomar.de)

**Abstract.**

European heat waves have increased during the two recent decades. Particularly 1994, 2015 and 2018 were characterized by a widespread area of cold North Atlantic sea surface temperatures (SSTs) as well as positive surface temperature anomalies across large parts of the European continent during boreal summer. The European heat wave of 2018 is further suggested to be induced by a quasi-stationary and high-amplified Rossby wave pattern associated with the so-called quasi-resonant amplification (QRA) mechanism. Another theory proposes, that certain heat wave events evolve through recurrent Rossby wave pattern, without involving stationarity. In this study, we evaluate whether and how the North Atlantic SST anomalies are related to European heat waves by using the ERA-5 reanalysis product.

A composite study reveals that cold North Atlantic SST anomalies favour a more undulating jet stream and a preferred trough-ridge pattern in the North Atlantic-European sector. Further we found that cold North Atlantic SSTs are more likely to occur with a stronger double jet, particularly in the North Atlantic sector. A performed wave analysis covering two-dimensional probability density distributions of phase speed and amplitude suggests that cold North Atlantic SST events enhance the dominance of transient waves. In the presence of a trough during cold North Atlantic events, we obtain a slow-down of the transient waves, but not necessarily an amplification or stationarity. The deceleration of the transient waves result in a longer duration of a trough over the North Atlantic accompanied by a ridge downstream over Europe, triggering European heat episodes. These findings are more in line with the recurrent Rossby wave pattern theory. Although a given preconditioning with a double jet structure may also be subject to the onset of quasi-stationary and high-amplified waves, our study found no general relation between cold North Atlantic SST events and these waves.

[revised manuscript text omitted]

A different mechanism that does not require stationarity in the Jet stream undulations or wave properties is the recurrent Rossby wave pattern (Röthlisberger et al., 2019), by which travelling waves transiently and repeatedly amplify over a preferred position, leaving their imprint on sub-seasonal averages. This mechanism has been linked to the 1994 summer hot spell. Upper-tropospheric transient Rossby wave packets (RWPs) are also suggested to be a major driver of the heat wave occurrences in 2003 and 2010 (Fragkoulidis et al., 2018). These cases support that a regional scale jet waviness diagnostic is more appropriate for investigating a potential link to temperature extremes than a circumglobal one (Röthlisberger et al., 2016).

The motivation of our study was originally to bridge the work of Kornhuber et al. (2017) and Duchez et al. (2016) for the first time. However, we won't perform full QRA diagnostics, instead focusing on pre-conditioning (double-jet structure) and the wave property distributions, which shift very differently depending on the dominance of stationary and amplified waves as opposed to transient and travelling waves. This will allow us to discern which mechanism dominates our analyses composited by SST anomalies.

Section 2 will present the data and methods used in our study. In section 3 we examine the effects of North Atlantic ocean anomalies on the jet stream and European surface temperatures by using a composite analysis. Further we will provide a wave analysis with respect to different underlying North Atlantic SST conditions.

**2   Data and methods**

This study is based on the European Centre for Medium-Range Weather Forecasts (ECMWF) ERA5 reanalysis data (Hersbach et al., 2019). We use daily averages from 1979 to 2019. The 300hPa geopotential height and the 300hPa zonal and meridional wind component as well as the sea surface temperature (SST) and 2m-air temperature are studied. We use a horizontal resolution of 2.5° longitude × 2.5° latitude.

A major part of our study is based on composites of the North Atlantic SST, so we should consider that no potential underlying trend would lead to a bias in the results, compositing more extreme heat episodes occurring later in the studied period.

[Figure]

**Figure 1.** MJJA trend for SST (ocean) and 2m-Temp. (land) based on the ERA5 data set ranging from 1979 to 2019..

Therefore we removed the trend of the SST data, although the North Atlantic area is not captured by a significantly positive
90    long term trend as shown by the black box in Fig. 1. The long term trend is in general stronger over the continent. An enhanced
increase in SST and 2m-air temperature is further identified at higher latitudes due to the commonly known polar amplification
(Coumou et al., 2018).

**2.1  Double Jet Index**

The investigation of double jet structures is crucial, because it provides a necessary precondition for the onset of the QRA-
95    mechanism, which leads to Rossby-wave amplification and quasi-stationary conditions (Kornhuber et al., 2017). We divided
our analysis into two different double jet measures:

The first measure of the double jet is zonally restricted. Here, it is important to note, that our definition of the double jet
differs from the study made by Kornhuber et al. (2017). They used a circumglobal zonally averaged zonal wind for double jet
examination. With this first measure, we are not able to directly compare our values to those made by their study. However,
100   we define a double jet index (DJI) only in the North Atlantic sector for two main reasons: because a double jet predominantly
forms within this longitude range and not globally (Hall et al., 2014), and because we want to focus on the effect on European
temperatures, which could be considered as the downstream response of the North Atlantic upper-tropospheric state.

[Figure]

**Figure 2.** 300 hPa total wind speed of the climatological mean (1979-2019) for bimonthly averages (coloured) in ms$^{-1}$: a) May-June; b) July-August; solid connected grid points between 7 and 45°W are used for Double Jet Index calculation: a) for May and June; b) for July and August, respectively.

For this purpose, we produced bimonthly means of the climatological total wind speed of the 300hPa level for May and June as well as for July and August, respectively, visualized in Fig. 2. According to the climatological state, three individual
105 branches are established, covering the poleward and equatorward maximum and the minimum in between. In consideration of the characteristic Southwest-Northeast tilt of the jet stream in the Atlantic sector, we introduce a steplike detection with equal slopes throughout all branches and all months. These step-like branches are illustrated as black lines with grid points highlighted in Fig. 2.

In order to define the double jet for each month, we calculated grid-point averages in the longitudinal range of 7.5° to 45°W
110 based on the same climatological branches. For May and June the average is derived by using the values for the latitudinal ranges below:

    – 47.5° - 55°N for the poleward maximum U$^{north}$

    – 20° - 27.5°N for the equatorward maximum U$^{south}$

    – 32.5° - 40°N for the minimum in between U$^{mid}$

115 The seasonally varying structure of the jet stream climatology requires a latitudinal adjustment of the branches. For July and August the DJI is derived based on the same ranges except from the equatorward maximum, which is captured by a branch between 25° and 32.5°N. Note that, despite the difference in wind speeds, the position of the maxima-minima (solid-pointed

lines) in Fig 2 does not differ substantially, which is why bi-monthly averages are used instead of a more frequent one.

Averages for both maxima and the minimum in between are now applied for the calculation of the DJI, which mainly consists of the ratio between the DJI of an individual year t and the average of the DJI over all years (1979 - 2019, n = 41), where t = 1,..,n:

$$\text{DJI}_t = \frac{U_t^{north} + U_t^{south} - U_t^{mid}}{\left[ \frac{\sum\limits_{t=1}^{n} U_t^{north} + U_t^{south} - U_t^{mid}}{n} \right]}.$$

For comparison, a second measure is introduced to illustrate the double jet structure, This measure is based on the findings of Fig. 1 of the study by Kornhuber et al. (2017). They identified a double jet structure for the European heat waves in July 2003 and July 2010. Both double jets are shaped similarly with respect to the latitude. We visualize the circumglobal jet structure of the Northern Hemisphere represented by the zonal mean zonal wind for the case studies of European heat wave occurrences for JJA 1994, 2015 and 2018 (Fig. S1). In agreement with Kornhuber et al. (2017), we found a double jet structure, particularly during 1994 and 2018 with a similar latitudinal shape, supporting that we could define certain latitudinal ranges, where we identify the poleward and equatorward maximum as well as the minimum in between, as followed:

- $60° - 80°$N for the poleward maximum $U^{north}$

- $30° - 50°$N for the equatorward maximum $U^{south}$

- $50° - 70°$N for the minimum in between $U^{mid}$.

Latitude bands for the detection of the maxima are represented by black dashed lines, and the latitude band for the detection of the minimum is highlighted in cyan dashed lines in Fig. S1. The double jet is then calculated in the same way as previously shown for the locally restricted double jet measure: monthly averages of zonal mean maxima and minimum are together related to its climatology.

**2.2 North Atlantic SST composites**

The influence of North Atlantic SST anomalies on the jet stream and on European surface temperatures is evaluated by using a composite study. After removing the seasonal variability by using a daily-based climatology of the SST data, the average over a North Atlantic box (45 - 60° N, 15 - 40° W) is created, coinciding with the area used for the analysis made by Duchez et al. (2016).

Afterwards daily values of this North Atlantic SST box average between 1979 and 2019 are used for a selection of a lower and an upper quantile, representing cold and warm SST events. Additionally a reference is obtained by selecting the daily values between the 0.4 and 0.6 quantiles, delineating the medium SST composite.

The respective dates of cold, warm and medium composites are used to calculate the corresponding jet stream latitude over

the North Atlantic-European sector: the latitude of the wind speed maximum within 45 and 90°N will be detected for each longitude between 90°E and 90°W for the composited daily data. This detection is done for all three composites and they are then averaged over time, respectively.

Jet stream latitude anomalies are now defined as the difference between warm and medium as well as cold and medium SST
150    events, respectively. Cold and warm SST composites are represented by the lower quantile up to 0.25 and the upper quantile above 0.75, which is shown in Fig. 4a. Both composites are further produced for the lower quantile of up to 0.1 and an upper quantile above 0.9, illustrated in Fig. S2a.

**2.3 Wave diagnostics**

In order to diagnose the wave characteristics and figure out whether heat waves rather originate based on the QRA theory or
155    the recurrent and transient Rossby-wave activity, we examine two-dimensional probability density functions (PDFs) of phase speed and wave amplitude by using a Kernel-density estimate similar to Kornhuber et al. (2017). Wave amplitude and phase of each wave number at each time step and latitude within the selected range (37.5 - 57.5°N (Fig. 8, S4 (upper)) / 50 - 70°N (Fig. S4, S5 (lower))) were determined by applying a fast Fourier transformation (FFT) of the meridional wind field at 300hPa with respect to longitude. The calculation of the phase speeds deviates from the calculation made in the study by Kornhuber
160    et al. (2017). Here, phase speed is calculated as the difference between the phase transitions from one 12h time step to the next. Excluding a cubic-spline interpolation and doing a raw difference instead, suggests a simple mean of the corresponding amplitude between the two neighboring 12h values, as the phase speed is representative for the time in between the 12h interval. For the definition of troughs and ridges in the North Atlantic we use 300hPa GPH. A zonal mean $z_{zonal}$ for the latitude band 45 - 60°N and another value $z_{Atl}$ for the North Atlantic box (14 - 40°W, 45 - 60°N) are both calculated and then combined as:

165

$$z_{anom} = z_{Atl} - z_{zonal}.$$

This anomaly $z_{anom}$ denotes the deviation from the zonal mean over a certain location following the Eulerian framework. Due to conservation of angular momentum, we assume that an outstanding anomaly in a longitudinal band outside the North Atlantic box is compensated by another anomaly of the same magnitude pointing in the opposite direction, so the zonal mean would average out local anomalies Thus, variations in the zonal-mean are typically orders of magnitude smaller than the eddy
170    terms avoiding a potential bias.

The chosen North Atlantic region is climatologically not represented by preferred higher or lower GPH values, as the region is part of the well-known southwest-northeast tilt, which is visible in the climatological state (Hall et al., 2017; Woollings and Blackburn, 2012).

We separate wave properties, comparing their kernel-density estimates when a trough or a ridge is present over the North At-
175    lantic, defined as days where $z_{anom} < 500$ gpm and $z_{anom} > 500$ gpm, respectively. Based on a cold (0.1 quantile) or a warm SST event (0.9 quantile) we perform the difference between troughs and ridges, shown in Fig. 9, S4 and S5. Fig. 5 shows the distribution of trough-ridge occurrence with different SST quantiles over the North Atlantic.

**3 Compositing with North Atlantic SST anomalies**

Low frequency variability associated with the North Atlantic ocean plays an important role for Northern Hemisphere climate (Kim et al., 2018). In this section implications of North Atlantic ocean anomalies are studied, particularly during boreal summer. In section 3.1 we focus on the investigation of the relationship with European surface temperatures. The section 3.2 contains the examination of jet stream properties with an emphasis on the behaviour in the North Atlantic-European sector. North Atlantic SST composites are further used for an evaluation of wave parameters, performed in section 3.3.

**3.1 Relation with European surface temperatures**

Previous studies suggested that the Atlantic Ocean is an important driver of summertime climate over Europe (Sutton and Hodson, 2005; Cassou et al., 2005; Duchez et al., 2016). The North Atlantic Ocean was characterized by a widespread area of negative anomalies during the summer 2015 and is suggested to be crucial for the development of the European heat wave (Duchez et al., 2016; Mecking et al., 2019). We investigate whether a similar picture evolved during the heat wave in 2018.

[Figure]

**Figure 3.** Detrended JJA anomalies for SST (ocean) and 2m-air temperature (land): a) 2018 anomalies; closed boxes are used for 2015 anomalies according to Duchez et al. (2016); dashed boxes indicates the area of largest 2018 anomalies; b) seasonal means (JJA) between 1979 and 2019 produced by box averages; years of heat wave occurrences are highlighted.

Therefore we assemble a summer (JJA) mean of 2m-air temperature anomalies on land and SST anomalies over the ocean, where the long term trend (1979 - 2019) is removed, visualized in Fig. 3a. Closed boxes refer to the area used by Duchez et al. (2016), whereas dashed boxes capture the regions where 2018 anomalies were centered.

In summer 2018 the North Atlantic ocean exhibited SST seasonal mean anomalies of up to -2.5°C, whereas Europe experienced 2m-air temperature seasonal mean anomalies of about +2°C. Negative SST anomalies are slightly shifted towards the Northwest compared to 2015, corresponding to a displacement of the box for largest anomalies in summer 2018. Positive surface temperature anomalies are mainly spread over central Europe during summer 2015 (Duchez et al., 2016). Instead, summer 2018 shows an extension of positive temperature anomalies towards Scandinavia, justifying a stretching of the 2018 box towards higher latitudes. However, the overall SST anomaly patterns and European temperatures in 2018 resemble those in Duchez et al. (2016) for the summer 2015 (see their Fig. 1a), with cold anomalies in the North Atlantic and warm anomalies over Europe, although in slightly displaced regions.

Figure 3b explores whether the pattern seen in the summers 2015 and 2018 is seen in other years as well and shows time series of yearly JJA anomalies for the four different boxes from Fig 3a. Beside 2015 and 2018, the European heat wave in 2003 shows a slight anti-correlation between SST and 2m-air temperature. By contrast, the heat wave in 2010 does not reveal an anti-correlation between anomalies of both parameters. But the hot spell during 2010 was located further eastward with its center roughly at Moscow (Barriopedro et al., 2011).

The two box averages based on anomalies in 2015 and 2018 yield only slight differences in 2m- air temperature during the heat waves 2003 and 2010. The average over the 2018 box (dashed line) produced even negative surface temperature anomalies during summer 2015, as the heat wave was constrained to central Europe. By contrast, the temperature anomalies are enhanced for summer 2018, consistent with a widespread warming that extends towards Scandinavia (Fig. 3a). Although we can see a sensitivity of the 2m-air temperature values with regard to the different boxes used for the averages, we can overall summarize that the major heat waves over central Europe (1994, 2003, 2015 and 2018) are all showing the same feature during summer season (JJA): mostly strong negative North Atlantic SSTs are co-occurring with positive 2m- air temperatures anomalies. Therefore we are interested to make an analysis about how these parameters could be linked together in general and in particular during the summer seasons of European heat wave occurrences.

Previous studies suggest that negative anomalies of the North Atlantic SST are favorable for a phase-locked meander in the jet stream (Duchez et al., 2016; Mecking et al., 2019). Next, we investigate whether these anomalies are linked to a certain jet stream pattern, that triggers positive European surface temperature anomalies.

**3.2 Relation with jet stream properties**

In order to evaluate a link between North Atlantic ocean anomalies and the jet stream behaviour, we perform a composite study. Fig. 4a illustrates the behaviour of the jet stream maximum during the 25% coldest (blue) and warmest (red) North Atlantic SST days. The zero line depicts the case of the reference, the medium SST events, and the dashed lines represents the standard deviation of the mean (medium SST events). Significant values are marked as thick lines. For detailed information about the composition we refer to the section 2.2.

Variations in the jet stream maximum between cold and warm composites are continuous over the North Atlantic: the jet stream maximum based on cold SST events shows a southward displacement with a local minimum of about -1.5° in the range of 30 to 50°W, corresponding to a preference of a trough in this region. The jet stream based on warm SST events reveals a slight northward displacement west of 20°W.

Further downstream (20 - 40°E) the signs of both composites are mostly reversed. While we observe a rather small northward displacement for the cold composite sluggishly exceeding the standard deviation of the mean, the warm composite reaches deviation values of -2° relative to the medium events. Between 50 and 90°E the composites exhibit similarly shifted jet stream maxima.

A reduced composite size by selecting only the 0.1 and 0.9 quantiles for cold and warm composites, respectively, yields a similar picture with an even more undulating jet stream for the cold composite (see Fig. S2a in the supplement). Particularly the southward shift of the jet stream in the North Atlantic as well as over Asia is more pronounced with values of less than -2° in latitude.

In summary, cold SST events generally show a well-developed trough-ridge system over the North Atlantic-European sector. Warm SST events show the opposite behaviour with an overall similar amplitude.

During our case studies of summer seasons in 1994, 2015 as well as 2018 cold SST anomalies were present in the North Atlantic region (Fig. 3b). The behaviour of the jet stream during those periods is displayed in Fig 4b, c and d.

During summer 1994 and summer 2015, the jet stream maximum (solid line) runs continuously south of the climatological jet stream position (medium SST composite, dashed line). Nonetheless a stronger southwest-northeast slope, reaching from the Northeast-Atlantic until the local maximum in the northwestern tip of Russia, intensifies the trough-ridge pattern in the North Atlantic-European sector during both summer seasons. The strengthening of the North Atlantic trough is supported by a pattern of negative and easterly adjacent positive meridional wind anomalies of 4-6 ms$^{-1}$ in this sector (coloured).

The summer 2018 reveals an even stronger picture: the jet stream maximum shows a stronger than usual trough over New-foundland at 50°N and a ridge extending up to the northern tip of Scandinavia at around 70°N. These results are in agreement with Kornhuber et al. (2019), who illustrated significant and widespread meridional wind anomaly patterns, which support the exceptionally strong waviness of the jet stream, appearing in the North Atlantic-European sector. Near-zero phase speeds and a preferred phase position of the wave, consisting of a trough over the North Atlantic and a ridge over the British Isles, are suggested to essentially contribute to the generation of the European heat wave occurrence in summer 2018.

In Fig. S2b, we show the jet stream behaviour for summer 2003. This jet stream maximum as well as the meridional wind anomalies are in quite good agreement with the summers of 1994 and 2015. Compared to all the previously shown summer seasons, we observe a different structure in summer 2010 (Fig. S2c), as an example of underlying warm instead of cold North Atlantic SST's (Fig.3b): the jet stream maximum is generally shifted northwards, reaches its ridge maximum further eastward towards Asia and the meridional wind anomaly patterns change the sign throughout all longitudes observed.

Both, composite and case studies verified that cold North Atlantic SST patterns could be associated with a preferred trough-ridge phase position of the jet stream in the North Atlantic-European sector. Haarsma et al. (2015) supports that a cold anomaly in the subpolar North Atlantic ocean induces a reduction of turbulent heat release by the ocean to the atmosphere. Further it

[Figure]

**Figure 4.** a) Composite of MJJA 300 hPa jet stream latitude (wind speed maximum between 45 and 90° N) anomalies for cold North Atlantic SST events (blue - 0.25 quantile) and warm North Atlantic SST events (red - 0.75 quantile) with respect to medium SST events (black); dashed lines delineate the standard deviation of the mean; b) MJJA 1994 300 hPa total wind speed maximum for each longitude (solid) and total wind speed maximum for medium SST events (dashed); shading indicates 300 hPa meridional wind anomalies (MJJA 1994); c) same as b) but here for MJJA 2015; d) same as b) and c) but here for 2018.

will favour a high pressure anomaly located downstream of the cold anomaly, roughly around the British Isles. Such a pattern further resembles the positive phase of the summer North Atlantic oscillation (summer NAO) (Folland et al., 2009).

260   Hereafter, we are inspired to make a more detailed analysis of the trough and ridge occurrence with respect to different North Atlantic SST conditions. We studied the probability of the trough/ridge amplitude for the North Atlantic (Fig. 5a) itself and for Europe (Fig. 5b). The trough-ridge occurrence is investigated in both regions relative to the zonal mean by a decomposition into different SST quantiles. Probabilities of a trough (negative values) or a ridge (positive values) present in the North Atlantic are determined using the 300 hPa GPH.

[Figure]

**Figure 5.** Probability of trough and ridge amplitude represented as the anomaly of daily 300 hPa geopotential height for a) the North Atlantic box (15 - 40°W and 45 - 60°N) and b) the European box (0-25°E°W and 45 - 60°N) relative to the zonal mean of the same latitudes; values are decomposed into the different North Atlantic SST box quantiles based on the calculation described in section 2.2.

265   The North Atlantic JJA climatology is characterized by a nearly symmetric distribution and a maximum, which is in the area of negative amplitude values, implying that the North Atlantic box has in general a very slight preference for a trough. Colder SST composites (the lowest quantiles) show a tendency to shifting the whole probability distribution towards negative values, i.e. deeper and more long-lasting troughs. The opposite happens with the warmer SST composites (higher quantiles) with a distribution shift towards positive values, but to a lesser extent.

270   These results are consistent with the jet stream maximum based on cold and warm North Atlantic SST composites shown in the previous Fig 4a: a strengthening of the trough during cold events and a weakening present during warm events support the contrary behaviour of the jet stream maximum between both composites within the North Atlantic box.

The European region indicates that cold North Atlantic SST composites are more likely in the presence of a European ridge

[Figure]

**Figure 6.** Lead-lag distribution (from -60 to +60 days) of PDF anomaly relative to the JJA climatology for composite values of 300hPa based on cold North Atlantic SST's (0.1 quantile): negative lead-lag values imply a leading jet stream state, whereas positive values indicate a leading North Atlantic SST.

and vice versa. The distribution for Europe does not provide the mirror image of the distribution shown for the North Atlantic
275 region in the upper panel of Fig. 5, however, these findings complement very well the jet stream position composites shown previously.

The cold North Atlantic SST anomaly could initiate a steady linear response in the atmospheric circulation consisting of the counteraction of warm-air advection towards the anomaly. This happens downstream of the SST anomaly within the western flank of the anticyclonic circulation (Hoskins and Karoly, 1981; Held et al., 2002). Consequently, our observed temperature
280 contrast (Fig. 3) as well as the trough-ridge pattern (Figs. 4 and 5) could be explained by an atmospheric response to the North Atlantic surface cooling.

Now after confirming a preference for a trough-ridge pattern during cold SST events, we try to address the temporal relation between the parameters in more detail aiming with a lead-lag correlation study. Fig 6 shows the distribution from -60 to +60

days lag for the PDF anomalies relative to the JJA climatology. The values are composited based on cold North Atlantic SST's (0.1 quantile). Negative values explain a leading of the upper-tropospheric state and positive values represent a leading North Atlantic SST.

Overall the distribution anomalies stay fairly stable from -60 to +60 lag, although with higher magnitudes near 0 and negative lags. There is no hint of SST's leading the atmospheric circulation response, if something slightly the opposite. This is valid for both regions, but with a stronger tendency for the North Atlantic region, i.e. PDF anomalies are approaching zero towards a leading relationship of North Atlantic SST's (positive lead-lag values).

Changing the analysis to values based on the 0.25 quantile, would give the same distribution, but with lower magnitude, so qualitatively this result is not sensitive to the choice of quantiles (not shown). Consequently, we state that no conclusion can be obtained whether the North Atlantic SST or the jet stream is forcing each other. A possibility would be a two-way interaction or a feedback between both components, but this would require further study.

Next, we examine whether the preferred trough-ridge pattern in the North Atlantic-European sector evolves through of a certain mechanism like the QRA mechanism or due to recurrent Rossby-wave patterns.

**3.3   Relation with double jet structure**

A pronounced double jet structure is the basis for the onset of the process of wave resonance. Kornhuber et al. (2017) claim that the majority of detected QRA events are related to double jet structures. The relationship between North Atlantic SST anomalies and the double jet index (DJI) is shown in Fig. 7, where the upper panel illustrates the connection to the locally restricted DJI and the lower panel describes the relation to a circumglobal measure based on zonal mean zonal wind. The SST values remain unchanged throughout Fig 7 a) and b). Further description of the derivation is provided in section 2.1. The analysis of the scatter plot is restricted to monthly values within the extended summer season (MJJA). Values of recent summers with European heat waves (1994, 2003, 2010, 2015, 2018) are highlighted within the plot.

The ocean's property of low frequency variability is responsible for the persistence of SST values within a certain range throughout one individual summer. For instance, the summer SST anomalies in 2018 are constrained between -1.1 to -0.5°C.

Despite a strong scattering, the tendency of a more intense double jet pattern during cold SST anomalies is evident, supported by the tilted line of linear regression. We obtained a negative correlation coefficient for the relation to both DJI measures (North Atlantic: correlation: -0.2587 with a p-value: 0.0008; circumglobal: correlation -0.1590 with a p-value: 0.0420). As an example serves the case study of 2018, where a negative North Atlantic SST anomaly shows the higher possibility for a stronger double jet than usual. Particularly May 2018 stands out with a double jet pattern in the North Atlantic region, that is twice as strong as the climatology. Overall both measures give ta similar anti-correlation but with an enhancement for the relation to the locally restricted North Atlantic DJI, which is explained by a usual higher occurrence of a double jet within this sector, supported by the study of Hall et al. (2014). Thus, a precondition for a higher double jet, at least in the North Atlantic sector, could be more likely fulfilled by the establishment of cold North Atlantic SST values.

Once a double jet pattern is established, next stages of quasi-resonant amplification can develop (Kornhuber et al., 2017;

[Figure]

**Figure 7.** Scatter Plot of monthly mean values (MJJA) for the North Atlantic SST anomaly (15 - 40°W and 45 - 60°N) and a) DJI (North Atlantic), b) DJI (circumglobal); years of particular interest (1994, 2003, 2010, 2015, 2018) are denoted by 'month/year'; the black line represents a linear regression; for further description of the DJI we refer to section 2.1.

Coumou et al., 2018). Advanced analysis of the wave pattern is done by inspecting the wave amplitude and phase speed in the following section.

**3.4 Relation with wave parameters**

320 Here, we clarify whether the preference of an enhanced trough-ridge in the North Atlantic-European sector, that could lead to anomalous heat over Europe, is a result of wave resonance and amplification associated with the QRA-mechanism or related to a recurrent Rossby-wave packet activity. Detailed information about the methods is provided in section 2.3.

In Fig. 8 we produced the probability density functions of phase speed versus wave amplitude for the composited cold (0.1 quantile) and warm (0.9 quantile) North Atlantic SSTs. Anomalies (shading) are superimposed on the JJA climatology (contour lines).

[Figure]

**Figure 8.** Probability density function of phase speed versus wave amplitude (derivation described in section 2.3); anomalies are shown for the 10% coldest SST days (378 days) in the North Atlantic box (15 - 40°W and 45 - 60°N) (upper row) and for the 10% warmest SST days within the same North Atlantic box (bottom row); wave properties are shown for 37.5 - 57.5°N for comparison with Kornhuber et al. (2017); black contour lines denote the JJA climatology.

The JJA climatology features the preference for eastward-traveling waves (positive phase speeds) (Fig. 8). Quasi-stationary waves with a tendency of higher amplitudes as well as a fraction of westward-traveling waves are further evident from the spectrum, which is in agreement with the study by Kornhuber et al. (2017).

Assuming that QRA events are present, we expect positive PDF anomalies in the area of low phase speeds and high amplitudes and negative anomalies of low-amplitude fast waves. The study by Kornhuber et al. (2017) found these conditions in the spectra of wave numbers 6 to 8 and Kornhuber et al. (2019) explicitly show wave-7 activity during the European heat wave of 2018. We observe that cold SST events enhance the dominance of eastward-traveling waves and reduce the occurrence of quasi-stationary high-amplitude waves (Fig. 8, upper row), apparent throughout nearly all PDFs and particularly for wave number 5,7 and 8. A similar pattern but with lower magnitude is found by compositing the 0.25% coldest and the 0.25% warmest SSTs, respectively (not shown).

Another approach was made by compositing cold SSTs and high DJI values, selecting days that were in the lower 0.25 SST quantile and upper 0.25 DJI quantile simultaneously. Results are shown in Fig. S3 (upper row). The distribution of the anomalies yields a similar picture. These findings indicate no signature of quasi stationary and high-amplified waves by either composit-

340 ing cold SST events only or values during cold SST conditions, where additionally high DJI values are found.

Composites based on warm SSTs (Fig. 8, bottom row) only and together with low DJI values (Fig. S3, bottom row) exhibit generally a lower magnitude without an outstanding anomaly throughout all wave numbers. Based on these results, the previously identified trough-ridge pattern in the North Atlantic-European sector shown in Fig. 4 and Fig. 5 cannot be explained by a stationary wave pattern (Fig. 8 and Fig. S3), and thus we need to pay a closer look into the properties of transient and travelling

345 waves.

In order to probe the behaviour of transient waves preferred during cold North Atlantic SSTs, we take the 10% coldest SST events (378 days) and calculate the difference in wave phase-speed and amplitude PDFs while troughs (224 days) or ridges (35 days) are present over the North Atlantic shown in Fig. 9 (coloured). The PDFs hereby show a composite of wave properties for cold SST events, with a climatological view additionally (black contour lines).

[Figure]

**Figure 9.** Same as in Fig. 8 but here the 10% coldest SST days (378 days) in the North Atlantic box (15 - 40°W and 45 - 60°N) are selected and then separated whether trough (224 days) or ridge (35 days) are present over this box; anomalies (coloured) illustrate the difference between trough and ridge.

350

The anomalies are in general higher with respect to the previous ones shown in Fig. 8. Here, positive anomalies concentrate at near-zero phase speeds throughout all wave numbers, implying that with a trough present in the North Atlantic, travelling waves propagate slower during cold SST events and vice-versa: when a ridge passes over the cold SST anomaly in the North Atlantic, the wave's phase speed is faster.

355 This would result in a longer duration and a higher occurrence of a trough over the North Atlantic box during cold SST events. The remaining wave patterns even show an enhancement of low amplitude waves, which therefore cannot be classified as QRA events.

We found a similar result by using a different latitudinal section of 50 - 70°N, illustrated in Fig. S4. Note that anomalies shown Fig. 9 and S4 illustrate the difference between troughs and ridges, whereas the overall appearance combining them is provided

360 in Fig. 8. The relationship is not linear, as the counterpart, the 10% warmest SST events, does not show the opposite signature in trough-ridge differences (see Fig. S5). Additionally the anomalies are less pronounced during the warm SST composite.

Another mechanism is needed to explain the higher occurrence of troughs during anomalously cold SST's in the North Atlantic,

while the jet stream doesn't enter stationary conditions. These results rather merge with the findings made by Röthlisberger et al. (2019), who classified certain unusually long-lasting hot spells into the appearance of recurrent and transient Rossby wave packets, i.e. during the European heat wave in 1994. Davies (2015), Barton et al. (2016) and Fragkoulidis et al. (2018) explain in general that persistent surface weather can evolve when the synoptic-scale wave packets are structured in the way that individual troughs and ridges repeatedly occur at the same longitude, reinforcing hereby the synoptic-scale surface weather pattern in order to temporally extend their occurrence. The studies by Fragkoulidis et al. (2018) and Pilch Kedzierski et al. (2020) in particular showed that a sequence of Rossby wave packets were contributing to the excessive heat in western Russia during 2010 as well.

Although Fig. 9 does not illustrate an amplifying over a specific position, the transient and travelling wave slows down and spends relatively more time with a specific trough-ridge setting in the North Atlantic-European region. This does not exclude the possibility that some of these slower-moving trough-ridge systems over this sector become quasi-stationary or resonant in very specific events. But even if QRA ever kicks in, it doesn't change the fact, that climatologically, cold North Atlantic SSTs promote overall faster-moving waves, and not necessarily amplified (Fig. 8).

In this study, we were looking for the responsible Rossby waves that bridge the anti-correlation found between cold North Atlantic SSTs and European summer heat wave events (Fig. 3). We found that cold North Atlantic SST's are associated with a preferred trough-ridge pattern in the North Atlantic-European sector (Fig. 4 and 5), without identifying a distinct sequence of responses (Fig. 6). By analysing which waves dominate the establishment of the trough-ridge setting, high DJI values (Fig. 7) would provide a necessary precondition for the development of wave resonance and amplification. But no signature for these characteristics appear in the phase speed versus amplitude PDFs shown in Fig. 8. We do not find higher amplitudes and stationary waves in cold SST only or combined with DJI composites. We explain the occurrence of the pattern in the North Atlantic-European sector by transient waves being slower (but not necessarily stationary or amplified) while a trough is present over North Atlantic cold SSTs, apparent from Fig. 9. The higher amount of time spent with this setting leaves its signature on the jet stream latitude composites (Fig. 4), which further could initiate higher temperature anomalies over Europe (Fig. 3).

A given DJI preconditioning in individual events may be a subject to the QRA mechanism. But in our study wave resonance and subsequent amplification is not found to be the dominant cause of the jet stream latitude anomalies related to anomalously cold North Atlantic SSTs. The study by Wolf et al. (2018) describes a connection between quasi-stationary waves and European heat waves, without requiring resonance. These quasi-stationary waves are further suggested to originate from preceding sea surface anomalies (Wolf et al., 2019). Our findings further generalize the results of Wolf et al 2019, by including the slow down of eastward travelling waves without becoming necessarily stationary, while a trough and cold SSTs are present in the North Atlantic, correspondingly with ridges present downstream in the European sector.

Several previous studies addressed the role of diabatic processes and associated latent heat release being important for locally modifying Rossby waves and the jet stream structure through acceleration of vertical motion (Wirth et al., 2018; Pfahl et al., 2015; Steinfeld and Pfahl, 2019). The enhancement of ascending motion could further play an important role for the formation and intensification of blocking events downstream (Steinfeld and Pfahl, 2019). Another study highlighted that a distinct latent heating upstream from an anticyclone could lead to a decrease of the phase speed of eastward propagating waves (Steinfeld

et al., 2020). The identified reduction of the phase speed during cold SSTs and a trough present in the North Atlantic found in our study could therefore also have a potential relationship to latent heat release, which needs to be addressed in a follow-up study. It would be interesting to see how these theories could generally apply to a slow-down of waves, without necessarily becoming a stationary pattern.

**4  Conclusions**

An increase in European heat waves during the $21^{st}$ century gained a special attention about the drivers of these anomalous conditions. The study by Duchez et al. (2016) proposed a relationship between significantly cold SST anomalies, found in the eastern North Atlantic, and the European surface temperature via Rossby wave propagation. The study by Kornhuber et al. (2017) suggested that heat waves are referred to the QRA mechanism associated with wave resonance and amplification. Another study related the appearance of certain heat waves to recurrent Rossby wave patterns (Röthlisberger et al., 2019). Our study combined these theories for the first time. We summarize our main findings as following:

1. European heat waves, particularly the occurrences during 2015 and 2018 both reveal a widespread area of negative SST anomalies across the eastern North Atlantic and positive surface temperature anomalies in central to northern Europe in agreement with Duchez et al. (2016).

2. The jet stream behaviour during cold North Atlantic SST events as well as during the case studies of 1994, 2015 and 2018 suggest a strong association of a certain undulation of the jet stream, with increased occurrence in the North-Atlantic European sector of a trough-ridge pattern.

3. Cold North Atlantic SST anomalies are related to increased double jet occurrence, a necessary precondition for wave resonance associated with the QRA mechanism. However our wave analyses with phase speed versus amplitude PDF's do not find any QRA imprint related to cold North Atlantic SSTs, compositing cold SSTs only or together with DJI values.

4. The main conclusion of our study is that cold North Atlantic SST events overall accelerate transient waves (Fig. 8, top row). When a trough is present over the North Atlantic cold SST anomaly, the wave speeds tend to be slower with a not necessarily higher amplitude. These conditions during cold North Atlantic SSTs are associated with an increased occurrence of an upper-level trough (Fig. 4), provoking a more persistent ridge downstream over Europe and initiating long-lasting heat conditions subsequently. Our results are in line with the recurrent Rossby wave pattern mechanism described by Röthlisberger et al. (2019).

5. The relationship is not linear, as the warm SST events do not show a reversed signal in terms of phase speed. Regarding the preference of a ridge in the North Atlantic during warm events, the magnitude is less pronounced compared to the case of cold SST events.

The contribution of the SST to variations in the atmospheric state are not only important in the year-to-year variability, but also on a longer time scale, since global warming will increasingly leave its mark: understanding the atmospheric effect of the North Atlantic warming hole (Drijfhout et al., 2012) as a long-term negative SST anomaly seems to be essential for the summertime circulation and the potential development of European heat waves.

*Author contributions.* JK performed the double jet index (DJI) analysis, produced Figs. 1 to 4 and Fig. 7 and wrote the manuscript; RPK assisted in the methodology development, designed the wave analysis, produced Figs. 5,6, 8 and 9, commented on and rewrote parts of the manuscript; KM motivated the study, supervised its development and commented on the manuscript; KB contributed with ideas and suggestions.

*Competing interests.* The authors declare that they have no conflict of interest.

*Acknowledgements.* We thank the European Centre for Medium-Range Weather Forecasts (ECMWF) data server for the freely available ERA-5 reanalysis data.